# The role of higher-order thalamus during learning and correct performance in goal-directed behavior

**Danilo La Terra[1], Ann-Sofie Bjerre[1], Marius Rosier[1,2], Rei Masuda[1], Tomás J Ryan[1,2,3], Lucy M Palmer[1]\***

[1]Florey Institute of Neuroscience and Mental Health, University of Melbourne, Melbourne, Australia; [2]School of Biochemistry and Immunology and Trinity College Institute for Neuroscience, Trinity College Dublin, Dublin, Ireland; [3]Child & Brain Development Program, Canadian Institute for Advanced Research (CIFAR), Toronto, Canada

**Abstract** The thalamus is a gateway to the cortex. Cortical encoding of complex behavior can therefore only be understood by considering the thalamic processing of sensory and internally generated information. Here, we use two-photon $Ca^{2+}$ imaging and optogenetics to investigate the role of axonal projections from the posteromedial nucleus of the thalamus (POm) to the forepaw area of the mouse primary somatosensory cortex (forepaw S1). By recording the activity of POm axonal projections within forepaw S1 during expert and chance performance in two tactile goal-directed tasks, we demonstrate that POm axons increase activity in the response and, to a lesser extent, reward epochs specifically during correct HIT performance. When performing at chance level during learning of a new behavior, POm axonal activity was decreased to naive rates and did not correlate with task performance. However, once evoked, the $Ca^{2+}$ transients were larger than during expert performance, suggesting POm input to S1 differentially encodes chance and expert performance. Furthermore, the POm influences goal-directed behavior, as photoinactivation of archaerhodopsin-expressing neurons in the POm decreased the learning rate and overall success in the behavioral task. Taken together, these findings expand the known roles of the higher-thalamic nuclei, illustrating the POm encodes and influences correct action during learning and performance in a sensory-based goal-directed behavior.

**\*For correspondence:** lucy.palmer@florey.edu.au

## Editor's evaluation

The thalamus is the hub connecting sensory inputs to cortical processing. The elegant study here used 2-photon calcium imaging and behavioral tasks to reveal a role for the posteromedial nucleus of the thalamus in goal directed forepaw behaviors in mice.

## Introduction

Goal-directed behavior is crucial for survival in a dynamic environment. It involves the encoding and integration of sensory information that leads to specific rewarded behaviors (*Kepecs et al., 2008*; *Li et al., 2015*; *Takahashi et al., 2016*; *Xu et al., 2012*). This process must be dynamic, as flexible switching of learnt behaviors is required throughout life. The thalamus is a fundamental hub for the transfer of sensory information to the cortex, sending and receiving widespread innervation from numerous cortical and subcortical structures (*Oh et al., 2014*; *Sherman and Guillery, 1996*). Despite being positioned to coordinate the relay and integration of sensory information required during

sensory-based behavior, historically, the thalamus has been viewed as a passive sensory relay center with negligible contribution to higher-order brain function and behavior. Recent studies have challenged this classical view, illustrating the thalamus plays crucial roles in cognitive tasks such as attention (*Schmitt et al., 2017*; *Wimmer et al., 2015*; *Zhou et al., 2016*), sensory perception (*Saalmann and Kastner, 2011*; *Wilke et al., 2009*), motor preparation and suppression (*Casagrande et al., 2005*; *Yu et al., 2016*), cortical plasticity (*Audette et al., 2019*; *Gambino et al., 2014*), and learning (*Williams and Holtmaat, 2019*).

The higher-order thalamus is an enigmatic class of nonspecific (diffuse projecting) thalamic nuclei which send feedback input to sensory cortical areas (*Sherman and Guillery, 1996*). These thalamic nuclei are thought to play an important role in behavioral flexibility (*Wimmer et al., 2015*) as reported changes in firing patterns within the higher-order thalamus (*Ramcharan et al., 2005*; *Urbain et al., 2015*) may underlie cortical state changes during adaptive behavior (*Bruno and Sakmann, 2006*; *Poulet et al., 2012*). Specifically, the posteromedial nucleus of the thalamus (POm) is the higher-order thalamic nucleus subtending sensory processing in the primary somatosensory cortex (S1) (*Deschênes et al., 1998*; *Jones, 2007*). Sending dense projections to layers 1 and 5 of S1 (*Meyer et al., 2010*), the POm specifically targets a complex cortical microcircuit (*Audette et al., 2018*) which influences the encoding of somatosensory inputs (*Castejon et al., 2016*; *Mease et al., 2016*; *Urbain et al., 2015*; *Zhang and Bruno, 2019*) and decision-related information (*El-Boustani et al., 2020*). The POm is reciprocally connected with S1, but also receives and sends projections to secondary sensory, motor, premotor, and association cortices as well as many subcortical regions including the zona incerta and striatum (*Alloway et al., 2017*; *Oh et al., 2014*; *Trageser and Keller, 2004*; *Yamawaki and Shepherd, 2015*). Based on its influence on cortical sensory processing and known extensive connectivity, the POm may play an important role during learning and performance in behaviors which require both the perception and integration of sensory information, such as sensory-based goal-directed behavior. To test this, we used two-photon $Ca^{2+}$ imaging and optogenetics to investigate the role of POm projections in the forepaw S1 during learning, and chance and expert performance in tactile goal-directed behavior.

## Results

### $Ca^{2+}$ imaging of POm axonal projections in forepaw S1 during 'action' goal-directed behavior

To assess the activity of POm projection axons during tactile-based behavior, two-photon $Ca^{2+}$ imaging of POm axons projecting to forepaw S1 was performed in mice (p50–70) previously injected with the $Ca^{2+}$ indicator GCaMP6f (AAV1.Syn.GCaMP6f.WPRE.SV40) into the POm (see Methods; *Figure 1—figure supplement 1*). Mice were then trained to associate forepaw tactile stimulation (200 Hz, 500 ms) with a reward in a goal-directed tactile detection task (see Methods; *Figure 1B*). If mice correctly responded by licking a reward port within 1.5 s after receiving the tactile stimulus, a sucrose water reward (10% sucrose water) was delivered. We refer to this behavioral paradigm as 'action' goal-directed task (action task). Mice rapidly learnt this task, taking on average 4.38 ± 0.37 days to reach expert level (>80% correct responses to tactile stimulation; *Figure 1—figure supplement 2*). Once expert, $Ca^{2+}$ transients were recorded from POm axons that project to layer 1 of the forepaw S1 (48 ± 6.8 µm from the pia surface; *Figure 1C, D* and *Figure 1—figure supplement 3*). POm axons were excluded if they had greater than 95% correlated activity with other axons within the POm (see Methods and *Figure 1—figure supplement 4*). During correct performance in the tactile goal-directed task, large $Ca^{2+}$ transients (>2 SD of the baseline fluorescence; see Methods) were evoked in 90% of POm axons. This task-evoked POm axonal activity was greater than tactile-evoked $Ca^{2+}$ activity in the naive state, with a significantly higher probability of evoking a $Ca^{2+}$ transient in POm axons during the tactile task (0.35 ± 0.03 vs 0.21 ± 0.02; p = 0.0007; *Figure 1E* and *Figure 1—figure supplement 5*). Once evoked, the amplitude of the $Ca^{2+}$ transients was not significantly different between naive and expert mice (p = 0.58; *F* test). To further assess the activity of POm axons during tactile goal-directed behavior, we categorized all axons according to their peak activity during baseline (-2 to -1 s prestimulus), stimulus/response (response; 0–1 s poststimulus), or reward (2–3 s poststimulus) epochs of the task (*Figure 1F*). Since during expert behavior, only a small portion of axons responded during the stimulus epoch alone (6%; see Methods), the stimulus and response epoch were

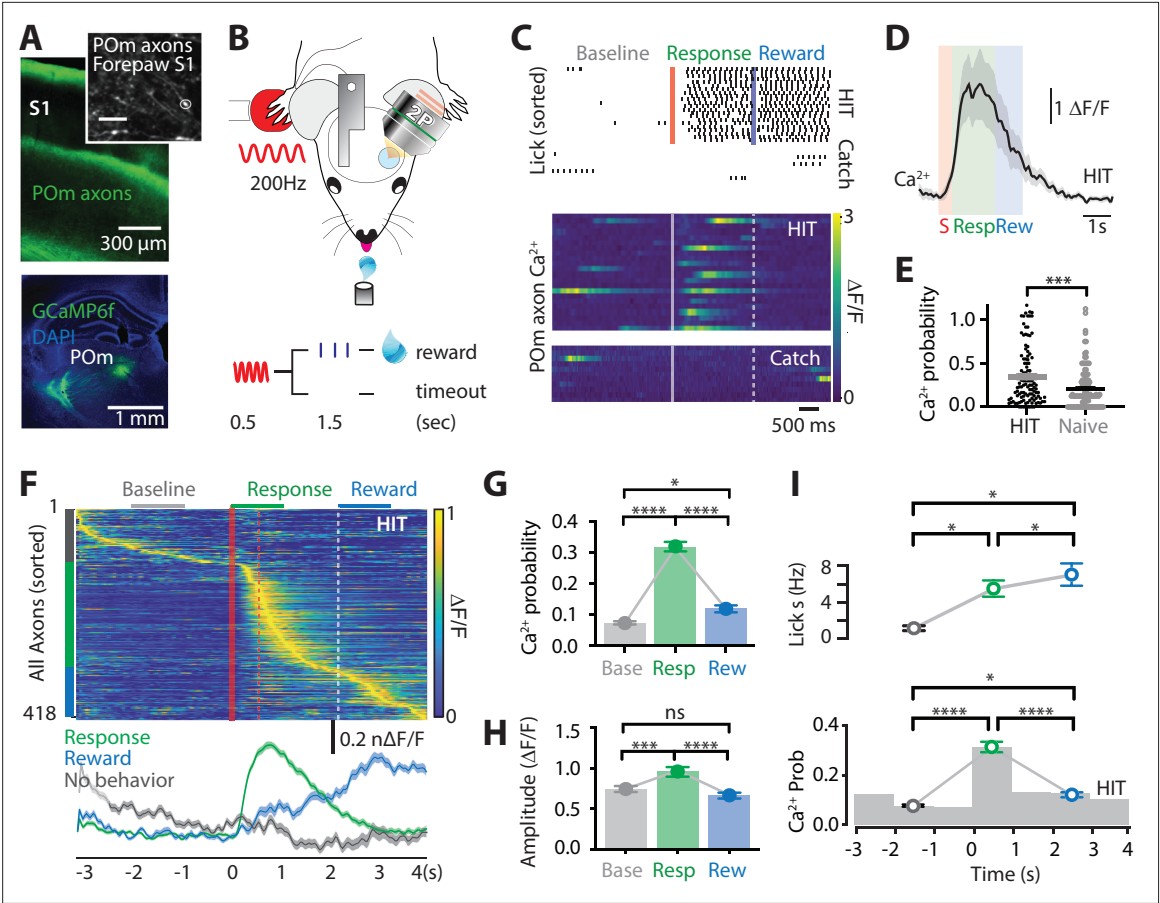

**Figure 1.** Ca²⁺ activity of POm axonal projections in forepaw S1 during tactile goal-directed behavior. (**A**) The Ca²⁺ indicator GCaMP6f was locally injected into the POm (bottom) which sends axonal projections to layers 1 and 5 of the forepaw S1 (top). Inset, in vivo two-photon Ca²⁺ image of POm axonal projections in forepaw S1 (depth, 60 μm; scale bar, 10 μm). (**B**) Two-photon Ca²⁺ imaging of GCaMP6f-expressing POm axons in forepaw S1 was performed in head-restrained mice trained to report the detection of a tactile stimulus (200 Hz, 500 ms) by licking a reward port. Correct responses (HIT) were rewarded with sucrose water reward (10 μl, 10% sucrose). (**C**) Top, raster plot showing a typical behavioral response (licks) sorted into correct HIT performance and Catch (no-stimulus) trials. Gray, spontaneous; red, tactile stimulus; green, response epoch; blue, reward epoch. Blue line, reward delivery. Bottom, example of Ca²⁺ activity pattern during correct performance and Catch trials from the POm axon in (**A**). Each row represents a single trial, sorted according to trial number. (**D**) Mass average with standard error of the mean (SEM; shaded area) of all stimulus-evoked Ca²⁺ transients in all axons during correct goal-directed performance (HIT; black). Behavioral epochs indicated by color bars (red, stimulus; green, response; blue, reward). (**E**) Probability of evoking a Ca²⁺ response during correct HIT behavior (black) compared with tactile-evoked activity in the naive state (gray, n = 113 axons; Mann–Whitney test). (**F**) Top, Ca²⁺ activity pattern during HIT performance in the tactile goal-directed task. Each row is an independent axon normalized to maximum fluorescence and sorted by the timing of the peak amplitude (gray, baseline; red, stimulus; green, response epoch; blue, reward epoch). Red lines, stimulus delivery. Dashed line, reward delivery. Bottom, average Ca²⁺ response in POm axons active during the stimulus and response epoch (green), reward epoch (blue); baseline (no behavior; gray). (**G**) The probability of a Ca²⁺ transient in POm axons during baseline (gray), response epoch (green), reward epoch (blue). n = 418 axons, 11 mice. Friedman test + Dunn's multiple comparisons test. (**H**) The amplitude of Ca²⁺ transients in POm axons evoked during baseline (gray), response epoch (green), and reward epoch (blue). n = 239 axons, 11 mice with evoked Ca²⁺ transients. Friedman test + Dunn's multiple comparisons test. (**I**) Top, average lick frequency during spontaneous (gray), stim/response (green), and reward (blue) epochs during correct HIT behavior. Bottom, histogram of Ca²⁺ transient probability in POm axons. *p < 0.05, ***p < 0.001, ****p < 0.0001.

The online version of this article includes the following figure supplement(s) for figure 1:

**Figure supplement 1.** Targeting and spread of AAV injections in the POm nucleus.

**Figure supplement 2.** Mice rapidly learn the tactile goal-directed task.

**Figure supplement 3.** AAV-mediated expression of ChR2-eYFP in the POm nucleus of the thalamus and its axonal projections in forepaw S1.

**Figure supplement 4.** Region of interest (ROI) selection and exclusion criterion.

**Figure supplement 5.** Tactile-evoked activity of POm axons projecting to forepaw S1 in naive and expert mice.

merged. Here, POm axonal activity was greatest during the response epoch, increasing signaling by more than four fold above baseline (probability per trial, 0.32 ± 0.02 vs 0.08 ± 0.003; $n$ = 418 axons, 11 mice; p < 0.0001; *Figure 1G*). POm axons also significantly increased activity above baseline during the reward epoch (probability per trial, 0.12 ± 0.007; $n$ = 418 axons, 11 mice, p < 0.0225). However, when compared with the response-evoked activity, active POm axons were reduced in number ($n$ = 275 vs 359 axons) and evoked rate (probability per trial, p < 0.0001), suggesting that POm axons preferentially encode the response epoch (*Figure 1G*). Direct comparison of the Ca²⁺ transient amplitudes from POm axons with both spontaneous and evoked activity ($n$ = 239 axons, 11 mice) illustrates that Ca²⁺ transients evoked during the response epoch (0.97 ± 0.04 $\Delta F/F$) were also significantly larger than transients evoked during both baseline (0.756 ± 0.02 $\Delta F/F$, p = 0.0005) and reward delivery (0.679 ± 0.0219 $\Delta F/F$; p < 0.0001), further highlighting the enhanced POm axonal signaling during the behavioral response (*Figure 1H*). Together, these results illustrate that the POm increases signaling in S1 during both response and reward delivery, with greatest activity during the behavioral response to tactile goal-directed behavior. Licking motion itself did not influence POm axon activity in forepaw S1, as there was no correlation between licking frequency and POm axonal activity (p = 0.923; *Figure 1I*). Furthermore, there was no detectable change from baseline in POm axonal Ca²⁺ activity during spontaneous licking (0.09 ± 0.027 Hz; p = 0.29; $n$ = 71 axons, 3 mice). Therefore, overall, POm axons in forepaw S1 encode response and, to a lesser extent, reward information during a tactile goal-directed task.

## POm axon activity in forepaw S1 is correlated with correct tactile goal-directed behavior

We next assessed whether POm axonal activity in forepaw S1 changes according to behavioral performance. Upon receiving a tactile stimulus in the action task, mice had to lick a reward port within 1.5 s to receive a sucrose water reward (HIT). However, if they did not respond during this epoch, then no water was delivered (MISS; *Figure 2A*). Despite performing at expert level, mice did not respond (MISS) to on average 12.63% ± 6.63% of tactile stimuli. To assess whether the activity of POm axons is also correlated with MISS behavior, evoked Ca²⁺ activity was directly compared in POm axons with both HIT and MISS activity ($n$ = 159 axons, 6 mice). Compared to correct HIT behavior, POm axons in forepaw S1 were overall less

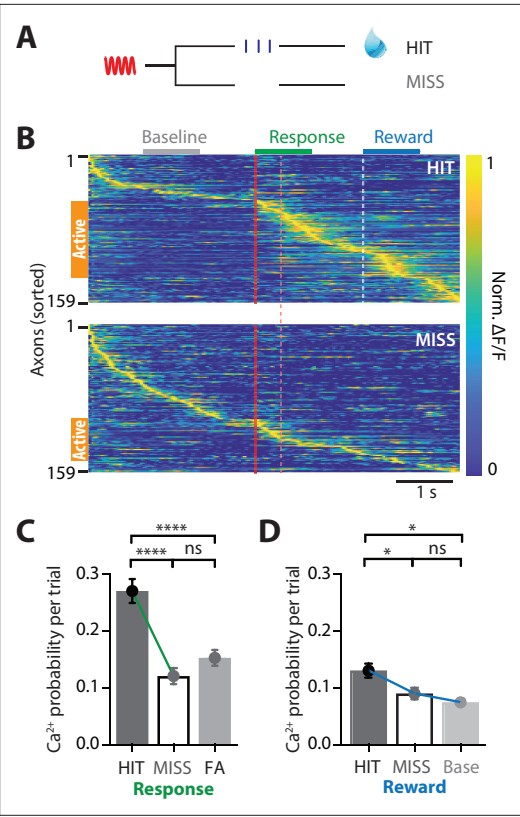

**Figure 2.** POm axonal projections in forepaw S1 have greatest activity during correct behavioral performance in a tactile goal-directed task. (**A**) Behavioral task design. Two-photon Ca²⁺ imaging of GCaMP6f-expressing POm axons in forepaw S1 was performed in head-restrained mice trained to report the detection of a tactile stimulus (200 Hz, 500 ms) by licking a reward port. Mice received sucrose water reward (10 µl, 10% sucrose) during correct responses (HIT), whereas incorrect responses (MISS) were unrewarded. (**B**) Ca²⁺ activity patterns in POm axons with Ca²⁺ transients evoked during HIT (top) and MISS (bottom) behavior during the tactile goal-directed task ($n$ = 159 axons, 6 mice). Gray, baseline; red, stimulus; green, response epoch; blue, reward epoch. Each row is an independent axon normalized to maximum fluorescence and sorted by the timing of the peak amplitude for both HIT and MISS trials. Orange bar denotes axons that were 'active' during the behavior. (**C**) The probability of a Ca²⁺ transient evoked during the response epoch in HIT (solid), MISS (empty), and false alarm (FA; dark gray) behavior. Wilcoxon matched-pairs signed rank test (HIT vs MISS) and Mann–Whitney test (FA vs HIT and MISS). (**D**) The probability of a Ca²⁺ transient evoked during the same time period as the reward epoch in HIT (solid), MISS (empty), and baseline (light gray). Friedman test + Dunn's multiple comparisons test. *p < 0.05, ***p < 0.001, ****p < 0.0001.

active during MISS trials (*Figure 2B*). In addition to an overall decrease in the number of axons active throughout the entire tactile goal-directed task (by 51%), there was also a significant decrease in the probability of evoking an axonal $Ca^{2+}$ event during the behavioral response in MISS trials (paired; HIT, 0.27 ± 0.02; MISS, 0.12 ± 0.01; p < 0.0001; *Figure 2C*). In contrast, the peak amplitude of the evoked $Ca^{2+}$ transients was similar during HIT and MISS trials (paired; response; HIT, 0.954 ± 0.08; MISS 0.888 ± 0.08; p = 0.470, *n* = 82 axons; reward; HIT, 0.635 ± 0.04; MISS, 0.708 ± 0.06; p = 0.252, *n* = 73 axons, 6 mice). Behaviorally speaking, HIT and MISS trials differ in the mouse movement, which has been shown to increase overall brain activity (*Stringer et al., 2019*). To investigate whether the increased POm axonal activity during the HIT response to tactile goal-directed behavior is due to body movement, we compared the evoked $Ca^{2+}$ activity in POm axons with catch trials where mice spontaneously licked for reward (false alarm, FA). Despite licking during FA trials, POm axons were significantly less active than during HIT trials (unpaired, probability per trial, 0.15 ± 0.01, *n* = 239 axons, 10 mice, p < 0.0001; *Figure 2C*). Since there was no significant difference between FA and HIT licking rates (p = 0.203, *n* = 9 mice), these data further suggest that POm axonal activity is not simply due to licking behavior. There was also a significant decrease in the probability of evoking an axonal $Ca^{2+}$ event during the reward epoch in MISS trials (HIT, 0.13 ± 0.02; MISS, 0.09 ± 0.01; p = 0.0408; *n* = 159 axons, 6 mice; *Figure 2D*). During MISS trials, POm axonal activity was similar to baseline rates (0.07 ± 0.007; p > 0.999; *n* = 159 axons, 6 mice; *Figure 2D*). Together, these data suggest that the POm encodes behavioral performance, increasing signaling between the POm and forepaw S1 during correct HIT behavior during both the response and reward epochs in a tactile goal-directed task.

## POm axon activity in forepaw S1 during suppression of a goal-directed action

Goal-directed behavior requires motor actions to be suppressed once they are no longer appropriate to achieve the current goal (*Jahanshahi et al., 2015*). To investigate the involvement of the higher-order thalamus during suppression of a previously learned goal-directed action, we performed $Ca^{2+}$ imaging from POm axons during a modified goal-directed paradigm. Here, mice previously injected with the $Ca^{2+}$ indicator GCaMP6f in the POm were trained in the 'action' goal-directed task (as above). Once expert (>80% correct responses to tactile stimulation), the behavioral paradigm was changed such that the mice only received the reward if they suppressed licking in response to the tactile stimulus (*Figure 3A*). We refer to this behavioral paradigm as 'action–suppression' goal-directed task (suppression task). To monitor cognitive arousal (*Bradley et al., 2008*), dynamic changes in pupil diameter were recorded while mice were performing the behavioral tasks. Despite the enforced behavioral (licking) suppression, the pupil diameter was significantly increased from baseline during the behavior (0.32 ± 0.05 to 0.44 ± 0.07; p = 0.031; *n* = 6 mice), indicating mice were engaged in the task. When compared with the action goal-directed task, there was no significant difference in peak pupil diameter during the pretrial baseline (0.32 ± 0.05 vs 0.29 ± 0.06 mm, p = 0.312), pretactile (0.35 ± 0.06 vs 0.31 ± 0.07 mm, p = 0.219), and post-tactile (0.44 ± 0.07 vs 0.41 ± 0.09 mm; p = 0.687; *n* = 6 mice; *Figure 3B*). Furthermore, there was no significant difference in correct performance rates during the action (83% ± 5% correct; *n* = 11 mice) and suppression (86% ± 7%; *n* = 6 mice; p = 0.57) tasks. POm projections in S1 were highly active during the suppression task, with evoked $Ca^{2+}$ transients that were significantly larger in amplitude than spontaneous activity (0.99 ± 0.06 vs 1.19 ± 0.06 ΔF/F; *n* = 144 axons, 6 mice; p = 0.0002). Similar to the action task, POm axons were most active during the response epoch in correct HIT trials (evoked rate, 0.24 ± 0.03; *n* = 144 axons, 6 mice; *Figure 3C*). Therefore, since the suppression task requires mice to suppress licking during the response epoch, the increased response activity was not correlated with licking behavior. To assess whether POm activity reflected behavioral performance in the suppression task, evoked POm $Ca^{2+}$ activity was directly compared during MISS trials. Similar to the action task, POm axons in forepaw S1 were less active during MISS behavior, with a significant decrease in the probability of response-evoked activity in MISS trials compared to HIT trials (HIT, 0.24 ± 0.03 vs MISS, 0.15 ± 0.03; *n* = 144/53 axons, 6 mice; p = 0.029; *Figure 3D*). Here, MISS behavior involves incorrectly licking for reward during the response epoch, further suggesting that POm axonal activity in S1 does not signal licking behavior. Taken together, in both the action and suppression tasks, POm axons in forepaw S1 preferentially encode the response epoch during correct performance (HIT trials). On average, the peak amplitudes (1.19 ± 0.06 vs 1.26 ± 0.05 ΔF/F, p = 0.3812) and durations (623 ± 50 vs 666 ± 35 ms; p = 0.2234) of $Ca^{2+}$

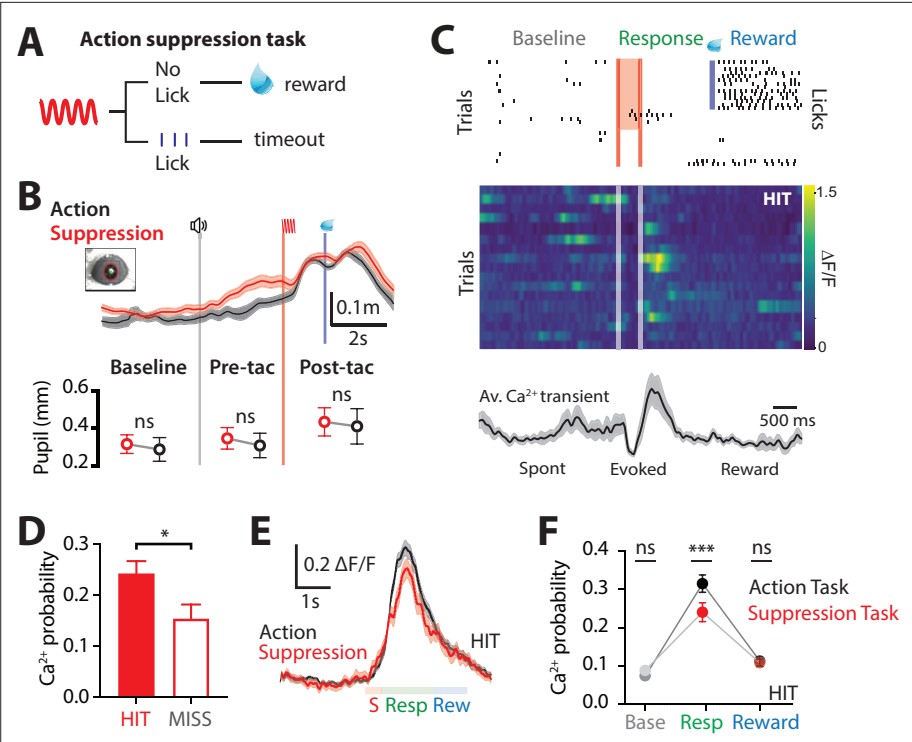

**Figure 3.** Ca²⁺ dynamics in POm axonal terminals during suppression of a goal-directed action. (**A**) Behavioral task design. Two-photon Ca²⁺ imaging of POm axon terminals was performed in head-restrained mice trained to suppress a previously learned goal-directed action. Mice were trained to withhold licking in response to forepaw stimulation (200 Hz, 500 ms) for 1.5 s to get a reward (10 µl, 10% sucrose water). (**B**) Top, average pupil diameter with SEM (shaded area) during correct performance in the 'suppression' goal-directed task (red) and 'action' goal-directed task (black). Bottom, comparison of pupil dilation during the 'action' and 'suppression' goal-directed tasks in baseline, pre-tactile stimulus (pre-tac) and post-tactile stimulus (post-tac) epochs (*n* = 6 mice: Wilcoxon matched-pairs signed rank test). Gray line, trial start; red line, stimulus, blue line, reward delivery. (**C**) Top, raster plot showing the typical licking response during correct performance of the task. Gray, spontaneous; red, stimulus; green, response epoch; blue, reward epoch. Blue line, reward delivery. Middle, Ca²⁺ activity pattern in an example axon during HIT trials. Bottom, average Ca²⁺ activity pattern with SEM (shaded area) in HIT trials for the example axon (*n* = 17 trials). Red line, stimulus delivery; blue line, reward delivery. (**D**) Probability of evoking a Ca²⁺ transient during HIT (correct suppression of licking behavior; red) and MISS (no suppression of licking behavior; red empty). *n* = 144 axons, 6 mice; Wilcoxon matched-pairs signed rank test. (**E**) Overlay of the mass average with SEM (shaded area) of the normalized Ca²⁺ activity pattern during correct performance in the suppression goal-directed task shown in (**C**) (red) and action goal-directed task (black). (**F**) Probability of evoked Ca²⁺ transients during baseline, response, and reward epochs in the 'suppression' goal-directed task (red) and 'action' goal-directed task (black). Mann–Whitney test. **p < 0.01, ***p < 0.001, ****p < 0.0001.

transients evoked during the action and suppression tasks were comparable (*Figure 3F*). However, during the suppression task, the probability of POm signaling during the response epoch was significantly decreased compared to the action task (p = 0.0007; *Figure 3F*). This contrasts with the similar probability of evoked POm signaling during the reward epoch (p = 0.87; *Figure 3F*). Together, these results further support the increased signaling of POm axons within forepaw S1 during correct (HIT) goal-directed active behavior.

## POm axon activity during switching of tactile goal-directed behavior

Flexibly switching motor actions in response to changing conditions is crucial for survival. Termed 'behavioral flexibility', this enables changes in the behavioral response to sensory information in dynamic environments. To investigate the role of POm during switching of rewarded behavior, we performed Ca²⁺ imaging from POm axons in forepaw S1 as mice transitioned from the 'action' goal-directed task to the 'action–suppression' goal-directed task (*Figure 4A*). We refer to this behavioral

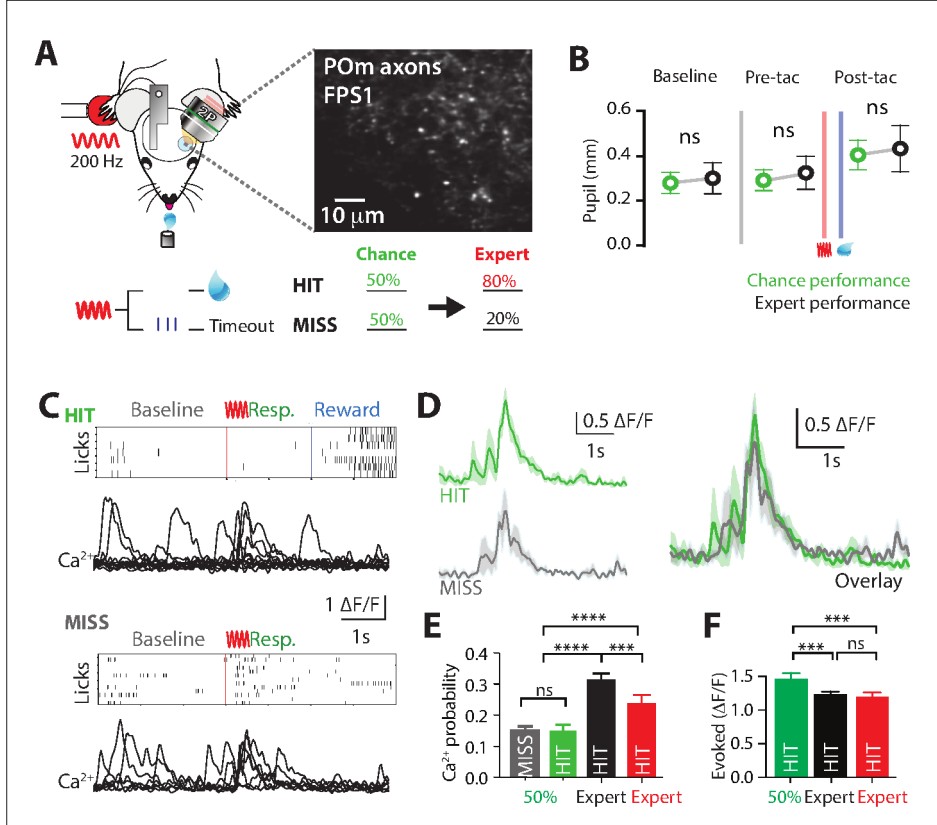

**Figure 4.** Ca²⁺ activity of POm axonal projections in forepaw S1 during chance performance and behavioral switching. (**A**) Behavioral task design. Ca²⁺ imaging from POm axons in forepaw S1 was performed as mice transitioned from the 'action' goal-directed task to the 'action–suppression' goal-directed task (50% correct performance, green). (**B**) Average pupil dilation during baseline, pre-tactile stimulation (pre-tac) and post-tactile stimulation (post-tac) during the 'switch' (green) and 'action' task (black; $n = 6$). Red bar, tactile stimulus; blue bar, reward delivery. (**C**) Example licking behavior and associated Ca²⁺ responses from an example axon during HIT (top) and MISS (bottom) trials. (**D**) (left) Individual and (right) overlay of average with SEM (shaded area) of evoked Ca²⁺ transients during correct (green) and incorrect (light blue) performance from example in (**C**). (**E**) The probability of a Ca²⁺ transient in MISS (gray) and HIT (green) trials during chance performance (gray), and expert HIT performance in 'action' (black) and 'suppression' (red) tasks. (**F**) Peak amplitude of evoked Ca²⁺ transients during HIT trials in the action (black), switch (green), and suppression (red) behavioral task. Error bars indicate the mean ± SEM. **p < 0.01, ***p < 0.001, ****p < 0.0001.

paradigm as 'switching'. On average mouse performance returned to chance level (50% correct performance) 2.25 ± 0.47 training sessions after switching the rewarded behavior. To monitor task engagement, pupil tracking was performed during the switch in behavior. Compared with correct performance in the active goal-directed behavior, there was no significant difference in pupil peak diameter during pre-trial baseline (0.28 ± 0.05 vs 0.30 ± 0.07 mm, p = 0.6871, $n = 6$ mice), pre-tactile (0.29 ± 0.05 vs 0.32 ± 0.07, p = 0.4372, $n = 6$ mice) and post-tactile epochs (0.40 ± 0.06 vs 0.43 ± 0.01; p = 0.6874; $n = 6$ mice, **Figure 4B**). Although equally engaged in the task, the activity of POm axonal projections in forepaw S1 was overall reduced during chance (50% correct) nonexpert behavior. Unlike expert behavior, the evoked rate of POm activity during chance performance did not reflect task performance, with similar evoked rates during both HIT (no lick, rewarded) and MISS (lick, unrewarded) responses (probability per trial, 0.15 ± 0.02 vs 0.15 ± 0.01, p = 0.74; $n = 121$ axons, 4 mice; **Figure 4C–E**). This rate of evoked activity during chance performance was similar to naive mice (p = 0.159), and significantly reduced when compared to expert performance (**Figure 4E**). Furthermore, during chance performance in nonexpert mice, POm projections in S1 did not signal correct performance nor reward delivery as there was no difference in the evoked rate of POm axonal Ca²⁺ activity during the behavioral response and reward epochs (probability per trial, 0.15 ± 0.02 vs 0.14 ±

0.02; p = 0.62; *n* = 121 axons, 4 mice). Taken together, unlike expert behavior, POm axonal activity in forepaw S1 was reduced and not correlated with the behavioral response during chance, nonexpert, performance in a goal-directed task.

To further investigate the potential role of the POm in behavioral switching, direct comparison of the amplitude of POm axonal transients was performed in mice which performed all tasks (Action, Switch, and Suppression tasks; *n* = 4 mice). Although POm axons were less active overall, when evoked, the amplitude of $Ca^{2+}$ transients evoked during HIT (Action, 1.19 ± 0.08 ΔF/F; Switch, 1.51 ± 0.09 ΔF/F, Suppression, 1.13 ± 0.09 ΔF/F; p = 0.0003; *n* = 77/69/47 axons, 4 mice; *Figure 4F*) and MISS (Action, 0.76 ± 0.05 ΔF/F; Switch, 1.37 ± 0.07 ΔF/F, Suppression, 1.02 ± 0.08 ΔF/F; p = 0.0001; *n* = 79/88/30 axons, 4 mice) performance during chance behavior was significantly larger than expert behavior. The lower evoked rate, but larger POm axonal transients during chance performance in a goal-directed task suggests a shift in the activity of POm input to forepaw S1 during nonexpert behavior, as mice are adjusting their behavioral strategy while learning the new goal-directed task.

## The influence of POm input during expert goal-directed behavior

The results above suggest the POm axonal activity in forepaw S1 is greatest during correct HIT behavioral response during expert, but not chance, performance in tactile goal-directed behavior. To investigate the role of this POm input on the correct performance during expert behavior, the POm was photoinhibited while expert mice performed the goal-directed task. Here, the inhibitory opsin, archaerhodopsin (ArchT; AAV1.CAG.ArchT.GFP.WPRE.SV40, 60 nl) was unilaterally injected into the POm. First, the effectiveness of 565 nm LED photoinhibition of POm neurons expressing ArchT was tested using patch clamp electrophysiology in the thalamic brain-slice preparation. Although photoinhibition did not completely abolish action potentials in POm neurons, the evoked firing rate was significantly decreased by 64% ± 13% (p = 0.031; *n* = 6 neurons; *Figure 5—figure supplement 1*). Next, we tested the influence of this decrease in POm activity on active goal-directed behavior in expert mice. A fiber-optic cannula was chronically inserted into the POm which was previously injected with ArchT (see Methods and *Figure 5A*) and mice were trained in the 'action' goal-directed task. Importantly, the duration of training and baseline performance was not affected by the injection of the inhibitory opsin into the POm (*Figure 5—figure supplement 2*). Once expert (>80% correct performance), the POm was initially photoinactivated with interleaved yellow LED light (565 nm, 5 mW, 2 s) during the stim/response epoch as this was when the POm was most active during the behavior (see Methods). Our findings illustrate that partial photoinactivation of the POm during the stimulus and response epoch produced a significant reduction in the overall behavioral performance (*d* prime, 2.58 ± 0.15 vs 2.23 ± 0.26; *n* = 9 mice; p = 0.04; *Figure 5B*) while no change was observed in the control group injected with green fluorescent protein (GFP) in the POm (*d* prime, 2.62 ± 0.24 vs 2.83 ± 0.29; *n* = 9 mice; p = 0.26; *Figure 5—figure supplement 2*). Specifically, the reduction in correct performance following POm partial photoinactivation was due to a significant decrease in performance during HIT trials (*z*-score, 2.09 ± 0.09 vs 1.77 ± 0.15; *n* = 9 mice, p = 0.02) and not the rate of FAs (*z*-score, 0.48 ± 0.16 to 0.46 ± 0.20; *n* = 9 mice, p = 0.82; *Figure 5C*). Despite this change in performance, POm photoinactivation during the stim/response epoch did not alter licking behavior as there was no significant difference in the latency to the first lick (control, 351 ± 29 ms vs ArchT, 342 ± 26 ms, *n* = 9 mice, p = 0.1282, *Figure 5D*). The specific influence of POm partial photoinactivation during the stim/response epoch is consistent with the increased signaling of POm axons within forepaw S1 during this epoch in expert mice (*Figures 2 and 3*). Since POm axons within S1 also increased activity above baseline during reward delivery, albeit less than during the behavioral response, we next tested whether photoinactivation of the POm during the reward epoch influenced behavioral performance. Here, no change was observed in overall behavioral performance when POm was photoinactivated during reward delivery (ArchT *d* prime, LED ON 3.54 ± 0.27 vs LED OFF 3.64 ± 0.09; *n* = 5 mice; p = 0.99; *Figure 5E*). There was also no change in overall behavioral performance during LED ON in reward delivery in control (GFP) mice (GFP *d* prime, LED ON 3.70 ± 0.19 vs LED OFF 3.69 ± 0.12; *n* = 6 mice; p = 0.75). Furthermore, similar to photoinactivation during the stim/response epoch, there was no influence on licking behavior when the POm was photoinactivated during the reward epoch (*Figure 5F*). Taken together, decreasing POm activity during the stim/response epoch in expert mice influenced correct performance in a goal-directed task, suggesting the higher-order thalamus specifically influences correct, but not incorrect, goal-directed responses.

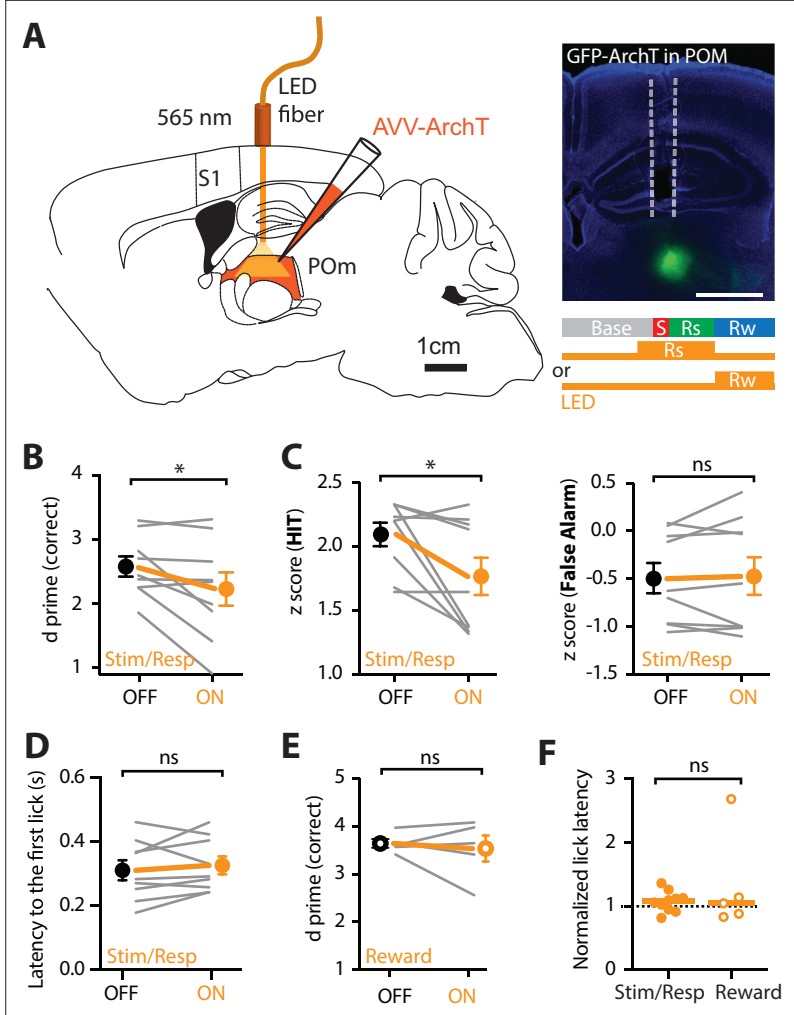

**Figure 5.** Optogenetic inactivation of the POm during an active goal-directed task. (**A**) Left, experimental design. The inhibitory opsin, archaerhodopsin (ArchT) was unilaterally injected into the POm and a fiber-optic cannula was chronically inserted into the brain. Right, localized ArchT spread in POm and fiber-optic track (dotted line), bar = 1 mm. POm was photoinactivated (590 nm, 5 mW, 2 s) either 500 ms prior to, and during the stimulus (S) and response (Rs) epochs (Stim/Resp), or during the reward epoch (Rw) in expert mice performing the 'action' goal-directed task. (**B**) Behavioral performance (*d* prime) for LED OFF vs LED ON during the stim/response epoch (*n* = 9 mice). Wilcoxon matched-pairs signed rank test. (**C**) *z*-Score during (left) HIT and (right) false alarm for LED OFF vs LED ON during the stim/response epoch (*n* = 9 mice). Wilcoxon matched-pairs signed rank test. (**D**) Latency to the first response lick in LED OFF vs LED ON during the stim/response epoch. Wilcoxon matched-pairs signed rank test. (**E**) Behavioral performance (*d* prime) during LED OFF and LED ON during the reward epoch in expert mice performing the 'action' goal-directed task (*n* = 5 mice). Wilcoxon matched-pairs signed rank test. (**F**) Normalized latency to the first response lick during LED ON in the stim/response epoch (solid) and reward (empty) epoch (normalized to the latency to the first lick during LED OFF). Line, median. Mann–Whitney test. Individual values are shown. *p < 0.05.

The online version of this article includes the following figure supplement(s) for figure 5:

**Figure supplement 1.** POm neurons are partially photoinhibited by 590 nm LED.

**Figure supplement 2.** LED in POm does not alter goal-directed behavior.

## The influence of POm input during learning of goal-directed behavior

Our findings suggest that the POm changes activity patterns from naive to expert performance, suggesting that the POm may play a role in learning of a sensory-based goal-directed task. To test this, a fiber-optic cannula was chronically inserted into the POm which was previously injected with ArchT and mice were trained in the 'action' goal-directed task. During each training session, the

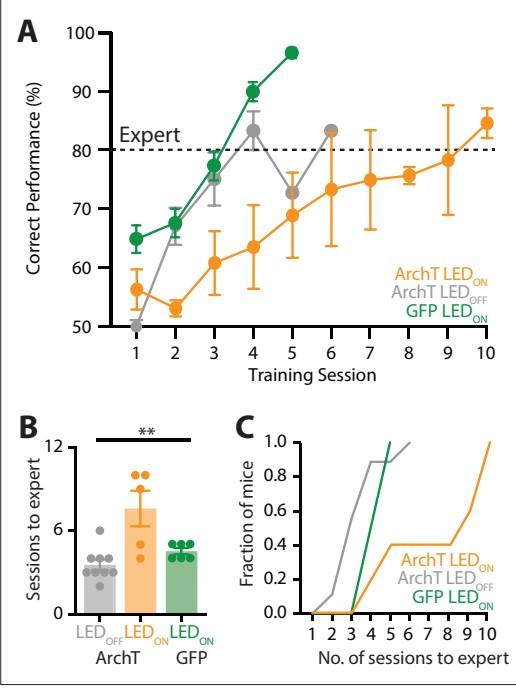

**Figure 6.** Optogenetic inactivation of the POm during learning of a goal-directed task. (**A**) Mice were injected with either a control AAV (GFP; green) or the inhibitory opsin, archaerhodopsin (ArchT) into the POm and trained in the 'action' goal-directed task. A fiber-optic cannula was inserted to the POm and LED (590 nm, 5 mW, 2 s) was either ON (ArchT, orange; GFP, green) or OFF (ArchT, gray) during all training sessions. Dotted line indicates expert (>80% correct) performance. (**B**) The number of training sessions required for mice to reach expert (>80% correct) performance in mice injected with ArchT (LED OFF, gray; LED ON, orange) or GFP (LED ON, green). Kruskal–Wallis test. (**C**) The number of sessions for mice to reach expert (>80% correct) performance. **p < 0.01.

POm was photoinactivated with yellow LED light (565 nm, 5 mW, 2 s) during the stim/response epoch as to not interfere with possible feedback pathways activated during reward, and this was also when the POm was most active during expert behavior. Here, when the POm was photoinactivated during learning, mice took on average 7.6 ± 1.3 sessions to reach expert (>80% correct) performance (n = 5 mice; *Figure 6A*). This is significantly greater than the number of sessions it took to reach expert performance in mice that were either previously injected with ArchT with LED OFF during learning (3.6 ± 0.4 sessions; n = 9 mice) or GFP (LED ON during learning; 4.5 ± 0.2 sessions; n = 6 mice; p = 0.004; *Figure 6B*). Therefore, decreasing POm activity during training in the goal-directed task influenced the rate of learning, with mice requiring more sessions to reach expert performance during POm photoinactivation (*Figure 6C*). Together, these findings suggest that the higher-order thalamus plays an important role in the learning of sensory-based tasks.

In summary, our findings suggest that POm axonal projections in forepaw S1 preferentially encode the behavioral response during learning and correct performance in tactile goal-directed behavior. Overall, POm axons were more active during expert performance, with greatest evoked rates in active behavior which required licking for reward, and POm photoinactivation during learning of the goal-directed behavior increased the number of training sessions required to reach expert performance. Taken together, these findings suggest that POm input to forepaw S1 shifts in strength and rate dynamically during learning and performance of a behavioral task, specifically encoding correct goal-directed action in the expert mouse.

## Discussion

The results presented here highlight the role of the POm during sensory-based goal-directed behavior. We used two-photon Ca²⁺ imaging to illustrate that POm axonal activity in forepaw S1 encodes correct behavioral response during expert performance in tactile goal-directed behavior. Specifically, POm axons increased activity in the response and, to a lesser extent, reward epochs during correct performance in expert behavior. This is in contrast to chance performance, where POm axonal activity did not correlate with task performance. Furthermore, the POm influences learning and performance in goal-directed behavior, as photoinactivation of archaerhodopsin-expressing neurons in the POm decreased learning rates and correct performance in expert behavior. Taken together, these findings illustrate that POm input to forepaw S1 specifically encodes correct performance during goal-directed behavior and influences sensory-based learning.

The POm is a higher-order nonspecific thalamic nucleus that is reciprocally connected with S1, but also receives and sends projections to motor, premotor, association cortices, and the brainstem (*Groh et al., 2014*) as well as many subcortical regions including the zona incerta and striatum (*Alloway et al., 2017*; *Oh et al., 2014*; *Trageser and Keller, 2004*; *Yamawaki and Shepherd, 2015*). Given the

extensive and heterogeneous organization of its afferent inputs, it is difficult to determine the input source driving POm activity during complex goal-directed behavior. However, since the POm receives input from the primary motor cortex (*Yamawaki and Shepherd, 2015*), it could be speculated that the increased axonal activity during the behavioral response has a motor origin. In fact, previous studies have shown that the thalamus is a circuit hub in motor preparation (*Guo et al., 2017*). While a general increase in POm activity has been reported during active states (*Urbain et al., 2015*), encoding of whisking related movement in the POm is relatively poor (*Moore et al., 2015*). In agreement, our findings illustrate that POm input in forepaw S1 does not specifically encode movement, as (1) POm activity is enhanced during the response epoch in both the action (licking) and suppression (no licking) tasks, (2) there is not a strong correlation between POm activity and licking frequency, and (3) POm activity is minimal during spontaneous licking and FA trials.

Here, we illustrate that POm activity is correlated with task performance in expert mice with greater signaling in POm axonal projections within forepaw S1 during correct HIT behavior in both the action and suppression tasks. Similar results were recently found in the thalamocortical circuit subserving the anterior lateral motor cortex (*Takahashi et al., 2021*) illustrating this may be a universal role of the thalamus. Although the POm encodes sensory information in naive mice, in the expert state, our findings suggest that this increased activity in POm axons during the response epoch is not primarily due to enhanced sensory encoding. Here, despite receiving exactly the same tactile stimulus, POm signaling in forepaw S1 is increased during correct HIT trials compared with MISS trials in both the action and suppression tasks. This difference in POm activity was not due to differences in licking behavior nor arousal, as POm activity was similar during the action and suppression tasks (which involved licking and not licking for reward) and did not reflect levels of arousal measured using pupil tracking. The difference in POm activity during the HIT and MISS trials was also not due to stimulus delivery as all experiments were monitored online via a behavioral camera to examine the location of the forepaw on the stimulus during all trials, and trials where the paw was not clearly resting on the stimulating rod were excluded from analysis. However, we cannot rule out that nondetectable changes in postures/paw grip may occur which may alter the effectiveness of the stimulus.

Although, overall, POm axons in forepaw S1 were predominantly active in the response epoch during correct performance in a tactile goal-directed behavior, the activity patterns of individual axons were heterogenous which may be due to a heterogenous population of POm neurons projecting to S1 (*Clascá et al., 2012*). Our findings illustrate that a subset of axons were correlated with the sensory stimulus and reward epoch during expert behavior. It is also possible that single POm axons may have heterogenous encoding which, since our findings are based on overall average activity per POm axonal projection, would not be evident in our study. Delving into encoding at the level of a single thalamic axon is an exciting direction for future research. How does POm encoding of goal-directed behavior compare to the activity of other thalamic nuclei which also project to forepaw S1? Of particular interest is the ventral posterolateral nucleus of the thalamus (VPL) which, in contrast to POm, targets the middle cortical layers of forepaw S1. Viewed as a feedforward (sensory) pathway, perhaps the VPL axons would be more active during the stimulus delivery and, in contrast to POm axons, their activity would be similar between the different behavioral tasks (action, suppression, and switch). It is of great interest to compare and contrast these different pathways to gain a holistic view of the role of the thalamus during goal-directed behavior, which will be the focus of exciting future studies.

In agreement with the POm axonal activity within S1, behavioral performance in expert mice was disrupted when the POm was photoinactivated during the stimulus and response epoch, but not during reward delivery. Together, these results suggest that the POm predominantly encodes the behavioral response. The behavioral effect was small which may be due to the following. Firstly, we illustrate that photoinhibiting LED light (565 nm) caused a significant decrease in the evoked action potential rate in POm neurons expressing archaerhodopsin in vitro. Taking into account the high firing rate of POm neurons in vivo, photoinhibition would not completely abolish POm activity in vivo. Therefore, during the goal-directed behavior, the POm is presumably still active, albeit at a reduced rate. Secondly, there was large behavioral variability which may reflect different rates of transfection and optical fiber placement. Thirdly, previous studies have illustrated that similar sensory-based goal-directed behaviors do not require primary cortical areas (*Hong et al., 2018*). Therefore, it is not expected that partially inhibiting an input stream to the forepaw S1 would have a large effect on the behavioral performance. Combined with the reported increase in POm axonal activity during

correct performance in expert mice, the influence of POm photoinactivation on task performance further supports the finding that POm encodes correct performance in goal-directed action. In this study, POm was also photoinactivated during learning of the tactile goal-directed behavior. Here, dampening POm activity significantly decreased the rate of learning, causing a greater than twofold increase in the number of training sessions required to reach expert performance. In our study, the influence of POm photoinactivation on goal-directed behavior was measurably greater during learning than expert performance, suggesting that the POm plays a vital role in learning. The role of the POm during learning requires more in-depth investigation, and since the POm targets many cortical and subcortical regions (*Alloway et al., 2017*; *Oh et al., 2014*; *Trageser and Keller, 2004*; *Yamawaki and Shepherd, 2015*), future studies with target-specific photoinhibition are required to illustrate which POm projection pathway specifically influences the learning and execution of goal-directed behavior.

In this study, considerable effort was made to ensure the specific targeting of POm. The POm was stereotaxically targeted with small volumes and the resulting fluorescence at both the thalamic injection site and the cortical layer targeted by the axonal projections was scrutinized after every experiment (*Gambino et al., 2014*). We note that our stereotaxic injections were not flawless and virus occasionally spread into ventral posterior nuclei, or along the injection pipette track and into high-order visual thalamic nuclei, superficial to the POm. If fluorescence was detected in nontargeted areas, then the experiments were excluded from analysis. It is possible that there was weak (undetectable) expression outside of the POm, however, these neighboring thalamic nuclei do not predominantly target layer 1 of the forepaw area of S1 (*Kamishina et al., 2009*; *Meyer et al., 2010*; *van Groen and Wyss, 1992*) and therefore would not significantly contribute to our calcium imaging findings. In the optogenetic photoinhibition experiments, targeting of the fiber-optic canula to the POm was confirmed after every experiment and weak expression of ArchT outside of the POm would therefore also have minimal impact on our findings.

To probe whether thalamocortical projections to S1 are dynamic and change activity patterns according to changes in reward expectation and delivery, we recorded the activity of POm axonal projections in S1 following a switch in the task contingency. In accordance with thalamic function playing an important role in behavioral flexibility (*Wimmer et al., 2015*), evoked axonal $Ca^{2+}$ activity was altered during the switch in rewarded behavior, suggesting a shift in action potential firing in POm thalamocortical neurons. Changes in firing mode have been reported in sensory higher-order thalamus of behaving rodents and primates (*Ramcharan et al., 2005*; *Urbain et al., 2015*) and may underlie cortical state changes during uncertain conditions and changes in reward expectation (*Bruno and Sakmann, 2006*; *Poulet et al., 2012*). These changes in firing patterns could drive different microcircuits (*Allen et al., 2017*; *Morgenstern et al., 2016*; *Tye and Uchida, 2018*) as POm inputs to the cortex directly target both excitatory and inhibitory neurons (*Audette et al., 2018*); however, more in-depth studies are required to directly investigate the influence of changing POm input on cortical microcircuits. Increased activity in the higher-order thalamus has also been associated with the expected value and significance of rewarded sensory stimuli (*Komura et al., 2001*) and may reflect learning-dependent strengthening of specific POm thalamocortical synapses (*Audette et al., 2019*). Our findings show that POm activity is enhanced during reward delivery in the tactile goal-directed task, although the absolute POm signaling is less than during the behavioral response.

Considering that patterns of cortical activity during behavior have been associated with task engagement, brain state, attention, motivation, or reward (*Kobak et al., 2016*; *Lacefield et al., 2019*; *Poort et al., 2015*; *Poulet and Petersen, 2008*; *Reimer et al., 2014*), we monitored pupil dynamics during the goal-directed tasks. We report that while overall POm activity increased concomitantly with pupil diameter during the behavioral response, this trend was reversed during reward delivery. By sorting POm axons according to their peak activity during the tactile goal-directed task, we revealed a subgroup of POm axons highly responsive during the reward epoch. This finding highlights the heterogeneity of the higher-order thalamus, with subsets of POm axonal projections specifically encoding either the stimulus, response, or reward delivery. In line with this finding, a recent report further supports the functional heterogeneity of POm cortical input and suggests it has a modulatory role in various brain regions during decision making in a goal-directed task (*El-Boustani et al., 2020*). However, overall, the results presented here illustrate that the POm predominantly transfers behaviorally relevant information to forepaw S1 during the response epoch of goal-directed behavior. Specifically, POm input to S1 is greatest in the response epoch during correct HIT performance in expert behavior. Although

this increased activity during the behavioral response epoch may not be necessary for maintaining the tactile information, these findings suggest that the POm does not simply encode sensory information, but it also reports behavioral outcome in learnt behavior and changes in behavioral state. Since the POm projects to various cortical and subcortical regions (*Oh et al., 2014*; *Yamashita et al., 2018*), the POm may also send task relevant information to other brain regions (*El-Boustani et al., 2020*). Likewise, since S1 also receives input from various brain regions, it would be of interest to investigate whether other input pathways send complimentary information during goal-directed behavior.

In summary, we show that the higher-order thalamus encodes correct performance during goal-directed behavior and influences the rate of learning. This finding expands the known roles of the higher-order thalamic nuclei, from sensory encoding to influencing learning and correct performance in goal-directed behavior. Overall, the thalamus is not a simple relay system. It encodes and influences learning of goal-directed behaviors which are crucial for survival in a dynamic environment.

# Materials and methods

All procedures were approved by the Florey Institute of Neuroscience and Mental Health Animal Care and Ethics Committee (17-091-FINMH) and followed the guidelines of the Australian Code of Practice for the Care and Use of Animals for Scientific Purposes.

## Mice

Wild type C57BL/6 female mice (PN30–80) were used in this study. Mice were housed in groups of six in a 12:12 natural light/dark cycle. All behavioral tests were performed during the light phase.

### Virus injection

All surgical procedures were conducted under isoflurane anesthesia (~1–2% in $O_2$). Body temperature was maintained at ~36°C and the depth of anesthesia was monitored throughout the experiment. Mice (~PN30–40) were placed in a stereotaxic frame (Narishige) and eye ointment was applied to the eye to prevent dehydration. The skin was disinfected with ethanol 70% and betadine before lidocaine (1%, wt/vol) was topically applied to the wound edges for additional local anesthesia. An incision in the skin (10 mm) was made to expose the skull and a small craniotomy (~0.5 × 0.5 mm) was made over the left posteromedial (POm) complex of the thalamus using the following stereotaxic coordinates: rostrocaudal (RC), 1.7 mm; mediolateral (ML), 1.25 mm; dorsoventral (DV), 3.00 mm from bregma. AAV1.Syn.GCaMP6f.WPRE.SV40 (Addgene plasmid # 100837, 1 x $10^{13}$ vg/ml) or AAV1.CAG.ArchT. GFP.WPRE.SV40 (Addgene plasmid # 29777, 1 x $10^{13}$ vg/ml) was slowly injected from a glass pipette (60 nl, Wiretrol, Drummond) for at least 5 min using an oil hydraulic manipulator system (MMO-220A, Narishige). The skin was then sutured and Meloxicam (3 mg/kg) was injected intraperitoneally (i.p.) for additional postoperative analgesia and anti-inflammatory action. Mice were then returned to their home cage for recovery.

### Chronic cranial window surgery

Mice previously injected with the $Ca^{2+}$ indicator GCaMP6f were anaesthetized (isoflurane, ~1–2% in $O_2$, vol/vol) and body temperature was maintained at ~36°C and the depth of anesthesia was monitored throughout the experiment. Eye ointment was applied to prevent dehydration and the top of the head was disinfected with ethanol 70% and betadine and lidocaine (1%, wt/vol) was topically applied for additional local anesthesia. The skin covering the skull was removed, and a craniotomy was performed over the left forepaw area of the primary somatosensory cortex (centered at coordinates: RC, 0 mm; ML, 2.3 mm; from bregma). The dura was left intact and a circular coverslip (3 mm diameter) was placed over the open craniotomy and seal attached to the skull with acrylic glue. A custom-made aluminum head bar (2 x 1 x 0.1 cm) was then attached to the skull for head-fixation using dental cement (C&B metabond, Parkell Inc). Meloxicam (3 mg/kg) was injected i.p. for additional postoperative analgesia and anti-inflammatory action. Mice were then returned to their cages to recover until behavioral training (~2 weeks).

### Habituation and behavior

Mice were trained to perform a goal-directed tactile task using a custom-made behavioral platform (*Micallef et al., 2017*). A 3- to 4-day habituation period preceded the beginning of the operant

conditioning. During this period, mice were handled and acclimatized to the behavioral setup. Mice were head restrained for incremental periods of time until habituated to head restraint. To maximize task engagement, a day prior to the beginning of behavioral training, mice were water restricted (1 ml/day of 10% sucrose water) and from this day onward this water regimen was maintained until the end of the experiment. Behavioral sessions lasted ~300 trials during which the mice typically obtained their daily water intake (1 ml/day) otherwise extra water was supplemented. $Ca^{2+}$ imaging was performed following this habituation phase for naive data.

## Behavioral platform

Mice were head-fixed to the recording frame and their paws rested unaided on either an active (contralateral) or inactive (ipsilateral) rod coupled to a stepper motor driven by an Arduino Uno microprocessor. The stepper motor delivered a pure frequency forepaw tactile stimulus (500 ms, 200 Hz). A water port was used to deliver a water reward (10 μl, 10% sucrose water) and licking frequency was recorded via a custom-made piezo-based lick sensor attached to the lick port. All behavioral tests were carried out in the dark while the animal behavior was monitored with an infrared sensitive camera (Microsoft lifecam). During the first training sessions, mice were habituated to tactile stimulus and reward delivery (typically one to two sessions). To establish an association between stimulus and reward, mice were able to self-initiate a trial by licking the water port which instantaneously triggered both stimulus and reward. After this habituation phase, operant conditioning was performed. **Action goal-directed task**: Background white noise (~40 DB) was played for the duration of each trial to indicate task onset and mask nontask-related sounds. Tactile stimulation (200 Hz, 500 ms) was delivered after a 3 s baseline period. Following stimulus presentation, mice were given a 1.5 s interval to report the detection of the tactile stimulus by licking the lickport (response epoch), after which reward was made available and cued by an auditory sound (400 Hz, 200 ms). Mice were then given a 2 s time window to retrieve the reward after which the trial terminated followed by an intertrial interval (ITI) of randomized duration (between 4 and 7 s). Only correct responses (licks during the response epoch) were rewarded (Correct) while failure to report stimulus detection was considered an incorrect response (Incorrect). Trials with no tactile stimulation (catch trials) were randomly interleaved with stimulus trials. Licking within the response epoch during a catch trial was considered a FA and punished with a timeout of incremental duration (2–7 s) while withholding licking was the correct response which was not rewarded, correct rejection (CR). Implementing catch trials and randomized ITI ensured that animals could not solve the task by adopting a time-based strategy. To facilitate learning, during the first training session the frequency of stimulus/catch trials was set to 90%/10%, respectively. The frequency of catch trials was progressively increased up to 40% and maintained at this ratio until mice could reliably perform at expert level (≥80 correct response rate). On average, mice reached expert level within 4.38 ± 0.37 training sessions. **Action–suppression goal-directed task**: Background white noise (~40 DB) was played for the duration of each trial to indicate task onset and mask nontask-related sounds. As in the action goal-directed task, tactile stimulation (200 Hz, 500 ms) was delivered after a 3-s baseline period. However, following stimulus presentation, mice were trained to withhold their licking for a 1.5-s interval. Mice were then given a 2-s time window to retrieve the reward after which the trial terminated followed by an ITI of randomized duration (between 4 and 7 s). Correct suppression of licking during this epoch was rewarded with sucrose water (10 μl, 10%). Conversely, if mice licked during this interval (early lick) no reward was delivered and the trial was aborted. Catch trials were used as in the action goal-directed task. Mice learned to reliably suppress licking (≥80% correct response rate) after an average 6 ± 0.85 training sessions (*Figure 5—figure supplement 2*). **Switchgoal-directed task:** Recordings were performed as mice transitioned from the goal-directed task to the action–suppression task. On average, mice expert in the action-task decreased performance to chance level after 2.25 ± 0.47 training sessions on the action–suppression task, at which point recordings were performed.

## Two-photon $Ca^{2+}$ imaging

Imaging of POm axons in forepaw S1 expressing the $Ca^{2+}$ indicator GCaMP6f was performed in awake behaving mice through a chronic cranial window approximately 3 weeks after virus injection. Head-fixed mice were placed under a two-photon microscope (Thorlabs A-scope) and POm axons located 48 ± 6.8 μm below the pia surface were excited using a Ti:Sapphire laser (Spectra Physics Mai Tai

Deepsee) tuned to 940 nm and passed through a 16x water immersion objective (Nikon, 0.8 NA). GaAsP photomultiplier tubes (Hamamatsu) were used for detection. The field of view (FOV) spanned 512 x 512 pixels and images were acquired at 30 Hz. To minimize photodamage, the excitation power was adjusted online to the minimal value sufficient to record $Ca^{2+}$ transients and the number of imaged trials for a given FOV was restricted to a maximum of 40. During each trial, animal behavior was monitored with an infrared sensitive camera (Microsoft Lifecam). Forepaw position on the tactile stimulator was recorded using an infrared webcam and analyzed post hoc. Due to the low resolution of the video recording, the video quality does not allow for detailed tracking of the paw, however, gross forepaw location on the tactile stimulator could be determined and any trials where the forepaw was not in contact with the stimulator were removed from further analysis.

## Cannula implant and photoinactivation of POm complex during learning and expert behavior

For optical inactivation of the POm complex, mice were injected ipsilaterally into the left POm with the inhibitory opsin AAV1.CAG.ArchT.GFP.WPRE.SV40 (60 nl; see virus injection). Following virus injection, a custom-made fiber-optic cannula (FT400EMT, 400 µm 0.39 NA, 2.5 mm fiber, Thorlabs) was slowly lowered down the injection track using a stereotaxic arm until the desired depth was reached (2.5 mm from pia). Dental cement (C&B metabond, Parkell Inc) was then applied around the edges of the cannula to secure it to the skull and left to dry for ~5 min. The same dental cement was used to attach a custom-made aluminum head bar (2 x 1 x 0.1 cm) to the skull for head-fixation. Meloxicam (3 mg/kg) was injected i.p. for additional postoperative analgesia and anti-inflammatory action. Mice were then returned to their cages to recover until behavioral training (~3 weeks). **Behavioral procedures:** After recovery, mice were trained on the action goal-directed task (see Habituation and Behavior). All behavioral procedures were performed using the Bpod behavioral platform (Bpod State Machine r1, Sanworks). **Photoinactivation:** Photoinactivation of the POm complex was achieved by delivering a light pulse (565 nm, 5 mW) through a 400 µm optical fiber (FT400EMT, Thorlabs) directly inserted into the cannula (FT400EMT, 400 µm 0.39 NA, 2.5 mm fiber, Thorlabs). A LED light source (LEDD1B, Thorlabs) coupled to a 565-nm LED filter (M565F3, Thorlabs) was used to generate the photostimulus. A custom-made light shield was placed over the animal's head to prevent scattered light from entering the animal visual field. Custom routines in Matlab were used to operate the behavioral platform and data acquisition. Photoinactivation was either performed during learning, or once mice reached expert level ( ≥ 80% correct response rate). During expert performance, the light pulse was delivered to inactivate the POm during the stim/response epoch (2-s duration; onset 500 ms prior to stimulus onset) or during the reward epoch (2-s duration; onset at reward delivery). During a typical experimental session (~300 trials), LED-ON and LED-OFF trials were randomly interleaved at a rate of 50% each. To photoinactivate the POm during learning of the goal-directed task, the LED was delivered during the stimulus and response epoch in all trials throughout learning (2-s duration; onset 500 ms prior to stimulus onset) until mice had reached expert performance or for a maximum of 10 consecutive days of training. For control experiments, mice were stereotaxically injected into their left POm (see virus injections) with AAV1-PAM MuseeGFP (kindly provided by Daniel Scott, 60 nl) and experiments carried out as above.

## Pupil tracking and analysis

To monitor engagement during the task, pupil tracking was performed in a subset of mice previously trained on the action goal-directed task for the ArchT experiments (see above). Pupil tracking was performed when mice were expert on both the action task and the action–Suppression task (see Habituation and Behavior). Pupil tracking was also performed during the transition between these tasks (switching) when their correct response rate dropped to chance level (~50%). Mice were head-fixed and the right eye illuminated with infrared light (850 nm LED, Thorlabs). This illumination did not affect pupil diameter. Behavioral sessions were performed on the same apparatus used for two-photon imaging inside an aluminum soundproof optical enclosure. However, some illumination (3.48 lux) was provided as we found that the pupil became maximally dilated and a-dynamic in complete darkness. An IR-sensitive camera (Basler aCA1300-200 µm) mounting a 50 mm lens (Kowa 50 mm/F2.8) was used to image pupil dynamics at 15 frames per second. Frames were triggered externally using an Arduino microprocessor connected to a Bpod (Bpod State Machine r1, Sanworks) which was then

used to operate the behavioral paradigm. Changes in pupil diameter were recorded and measured online using custom routines kindly provided by Bahr, Kremkow, Sachdev, and colleagues (*Bergmann et al., 2019*).

## Ex vivo whole-cell recordings and photoinhibition of POm neurons by ArchT activation

Mice (P40–45) previously injected with ArchT in the POm (>14 days prior) were anaesthetized with isoflurane (3–5% in 0.75 l/min $O_2$) before decapitation. The brain was then rapidly transferred and cut in an ice-cold, oxygenated solution containing (in mM): 110 choline chloride, 11.6 Na-ascorbate, 3.1 Na-pyruvate, 26 $NaHCO_3$, 2.5 KCl, 1.25 $NaH_2PO_4$, 0.5 $CaCl_2$, 7 $MgCl_2$, and 10 D-glucose (sigma). Coronal slices of the POm (300 μm thick) were cut with a vibrating microslicer (Leica Vibratome 1000 S) and incubated in an incubating solution containing (in mM): 125 NaCl, 3 KCl, 1.25 $NaH_2PO_4$, 25 $NaHCO_3$, 1 $CaCl_2$, 6 $MgCl_2$, and 10 D-glucose at 35°C for 20 min, followed by incubation at room temperature for at least 30 min before recording. All solutions were continuously bubbled with 95% $O_2$/5% $CO_2$ (Carbogen). Whole-cell patch clamp somatic recordings were made from visually identified pyramidal neurons using differential interference contrast microscopy. During recording, slices were constantly perfused at ~1.5 ml/min with carbogen-bubbled artificial cerebral spinal fluid containing (in mM): 125 NaCl, 25 $NaHCO_3$, 3 KCl, 1.25 $NaH_2PO_4$, 1.2 $CaCl_2$, 0.7 $MgCl_2$, and 10 D-glucose maintained at 30–34°C. Patch pipettes were pulled from borosilicate glass and had open tip resistance of 5–7 MΩ filled with an intracellular solution containing (in mM): 135 potassium gluconate, 70 KCl, 10 sodium phosphocreatine, 10 HEPES, 4 Mg-ATP, 0.3 $Na_2$-GTP, and 0.3% biocytin adjusted to pH 7.25 with KOH. Photoinhibition of POm neurons was achieved by shining a 565 nm LED light (1 s) onto the slice surface during somatic current injection steps (2 s). Firing rates before and during light application were quantified and compared to the same time period of the current step injection when no light was applied (*Figure 5—figure supplement 1*).

## Histology

At completion of each experiment, mice were transcardially perfused with phosphate buffer (0.1 M) and 4% paraformaldehyde (PFA) solution. Brains were collected and post fixed overnight (~12 hr) in 4% PFA at 4°C before being cut into 200 μm coronal slices using a vibratome (Leica VT1000 Automated Vibratome) and mounted on glass slides using mounting medium containing nuclear staining dye DAPI (Fluoroshild, Sigma). Images of the brain slices were acquired using wide-field fluorescent microscopy (Zeiss Axio Imager 2). Images were taken such that excitation light (EYFP, 555 nm; DAPI, 430 nm) was optimized below the maximum pixel saturation value for each fluorophore. To evaluate virus (GCaMP6f, ArchT) expression profiles in the POm complex, images of brain sections were registered to the corresponding coronal plates of the Paxinos mouse brain atlas (*Paxinos and Franklin, 2001*). Data from out of target injections or failed viral expression were removed from further analysis.

## Data analysis and statistical methods

### $Ca^{2+}$ data

All analyses were performed using ImageJ and custom written routines in Matlab or Python. Horizontal and vertical drifts of imaging frames due to animal motion were corrected by registering each frame to a reference image based on whole-frame cross-correlation. The reference image was generated by averaging frames for a given FOV in which motion drifts were minimal (< 15 pixels). Region of interests (ROIs) of axonal shafts or buttons were selected using the standard deviation of the entire imaging session (~6000–8000 frames) and manually drawn using the freehand tool in ImageJ. ROIs were selected so that each ROI represented a single POm axon. The activity profile was compared across all ROIs in a FOV. ROIs with similar activity profiles (where events were temporarily correlated in greater than 95% of trials) were presumed to be axonal branches or boutons of the same neuron and replicates were excluded from analysis. On average each FOV had 19 ± 2 ROIs. Across sessions the FOVs were overlapping, however, due to the size and shear density of axonal projections, individual axons were not imaged across sessions. To calculate the baseline fluorescence ($F_0$) for each ROI, first the average baseline florescence intensity (across 60 frames prior to stimulus onset, 2 s) of each trial was taken. Second, the rolling median of these average baseline values was measured and used as $F_0$. Fluorescence traces are expressed as relative fluorescence changes, $\Delta F/F = (F - F_0)/F_0$. Only $Ca^{2+}$

transients which were greater than 2x the baseline standard deviation ($F_0$ + (2x s.d.)) and above the threshold for a period longer than 200 ms were selected. ROIs were only considered for analysis if there was at least one Ca²⁺ transient reported during the trial (termed 'active axons'). The onset of a Ca²⁺ transient was defined as the time point at which a transient crossed the detection threshold ($F_0$ + (2x s.d.)). Both peak amplitude and probability of an evoked Ca²⁺ transient per trial were typically measured. Ca²⁺ transient amplitude may reflect the number of action potentials whereas Ca²⁺ transient probability is independent on the number of evoked action potentials. Average Ca²⁺ transient probability was measured as $\left(\left(\frac{\Sigma \text{ events}}{\text{time}}\right)/\Sigma \text{ trials}\right)$. The peak amplitude ($\Delta F/F$) was measured as the local maxima between the event onset and offset (i.e., when the falling edge of the transient crossed the threshold again). The duration (ms) of a Ca²⁺ transient was calculated as the time between the event onset and offset.

Three behaviorally relevant epochs were selected (1 s duration) for *spontaneous activity* (-2 to -1 s, relative to stimulus onset); for *response activity* (0 to +1 s, relative to stimulus onset) and for *reward activity* (0 to +1 s, relative to reward delivery). In a subset of axons (*n* = 107 axons, 3 mice), Ca²⁺ transients were further subcategorized as either occurring during the stimulus (0–500 ms) and response epochs (500–1000 ms) during the goal-directed task. Here, only a small portion of the axons (6%) were active only during the stimulus, whereas most axons were active during the response (only, 38% or combined, 56%). Therefore, to ensure accurate analysis of Ca²⁺ transients by using an expanded temporal window, the stimulus and response epoch were merged in the reported results. For probability comparisons, all ROIs were used, while only the subset of ROIs (i.e., axons) with detectable events (greater than the threshold) were used to measure amplitude and duration. This determines the difference in the number of axons used for each analysis. For direct comparison of POm activity during different epochs/behaviors, the subset of active axons with detectable Ca²⁺ events were typically used for analysis. On occasion, a mass average Ca²⁺ response was instead used, which is a mass average of all axons whether or not they had a response. Where appropriate, the variance of the peak Ca²⁺ amplitudes was compared using a *F* test. For displaying population activity, each row of the Ca²⁺ activity pattern is an individual axon, which is sorted by the timing of the peak amplitude for the particular behavioral condition.

## Pupil tracking

Videos of pupil tracking and animal behavior were acquired and checked post hoc to remove potential artifacts due to sudden eyelid closing. Analysis of pupil dynamics were performed using a custom written algorithm in python. Briefly, pupil tracking for the entire session was split into single trials (11 s duration) according to behavioral outcome. The average response profile was then calculated for each trial type for each mouse. Pupil dilation was monitored during a 4-s baseline period preceding the beginning of each trial. The average peak diameter was measured as the local maxima of the average pupil response during the baseline epoch (4 to 0 s, relative to trial start; baseline), pre-tactile epoch (-3 to 0 s, relative to stimulus onset, pre-tac), and post-tactile epoch (0 to +4 s relative to stimulus onset, post-tac).

## Behavior

The correct response rate was determined as *d* prime (the *z* transforms of HIT rate and FA rate $d' = z(H) - z(F)$) or as the fraction of correct trials over the total number of trials (HIT trials + correct rejection trials)/(stimulus trials + catch trials). The behavioral effects of POm photoinactivation were quantified by comparing correct responses of photoinactivation (LED-ON trials) vs. control (LED-OFF) trials, typically 150 each per experimental session. LED-ON trials and LED-OFF trials were randomly interleaved. The latency to first lick was calculated as the time of first lick occurrence after stimulus onset.

## Statistical analysis

No predetermined sample sizes were calculated prior to experiments. All statistics were performed using Prism software. The significance level was set at 0.05. Normality of all value distributions was assessed by Shapiro–Wilk test (α = 0.05). Standard parametric tests were used only when data passed the normality test (p > 0.05). Nonparametric tests were used otherwise. Only two-sided tests were used. Specific statistical tests used and sample sizes are shown in figure captions or text.

## Acknowledgements

We would like to thank members of the Palmer laboratory and Matthew Larkum for their helpful discussions and comments on the manuscript. We would also like to thank Verena Wimmer for her POm expertise and Ronny Bergmann, Viktor Bahr, Jens Kremkow, and Robert Sachev for use of their pupil-tracking software.

## Additional information

### Funding

| Funder | Grant reference number | Author |
|---|---|---|
| National Health and Medical Research Council | APP1130716 | Lucy M Palmer Tomás J Ryan |
| National Health and Medical Research Council | APP1063533 | Lucy M Palmer |
| National Health and Medical Research Council | APP1085708 | Lucy M Palmer |
| Australian Research Council | DP160103047 | Lucy M Palmer |
| Sylvia and Charles Viertel Charitable Foundation | | Lucy M Palmer |

The funders had no role in study design, data collection, and interpretation, or the decision to submit the work for publication.

### Author contributions

Danilo La Terra, Conceptualization, Data curation, Data curation, Investigation, Methodology, Visualization, Writing – original draft, Writing – review and editing; Ann-Sofie Bjerre, Investigation, Data curation; Marius Rosier, Data curation, Investigation; Rei Masuda, Data curation, Investigation; Tomás J Ryan, Funding acquisition, Supervision; Lucy M Palmer, Conceptualization, Funding acquisition, Methodology, Project administration, Resources, Supervision, Writing – original draft, Writing – review and editing

### Author ORCIDs

Ann-Sofie Bjerre http://orcid.org/0000-0001-6032-6502
Marius Rosier http://orcid.org/0000-0002-9732-5543
Lucy M Palmer http://orcid.org/0000-0003-3676-657X

### Ethics

All procedures were approved by the Florey Institute of Neuroscience and Mental Health Animal Care and Ethics Committee and followed the guidelines of the Australian Code of Practice for the Care and Use of Animals for Scientific Purpose.

### Decision letter and Author response

Decision letter https://doi.org/10.7554/eLife.77177.sa1
Author response https://doi.org/10.7554/eLife.77177.sa2

## Additional files

### Supplementary files
• Transparent reporting form

### Data availability

The source code for the behavioral system can be found online at https://github.com/palmerlab/behaviour_box, (copy archived at swh:1:rev:d4fad09624941bd2f25f9878c1ef304e84a6981a) as well as additional documentation at https://palmerlab.github.io. Calcium imaging data is available on Dryad https://doi.org/10.5061/dryad.1rn8pk0wb.

The following dataset was generated:

| Author(s) | Year | Dataset title | Dataset URL | Database and Identifier |
|-----------|------|---------------|-------------|-------------------------|
| Palmer L | 2022 | Data from: The role of higher order thalamus during learning and correct performance in goal-directed behavior | https://doi.org/10.5061/dryad.1rn8pk0wb | Dryad Digital Repository, 10.5061/dryad.1rn8pk0wb |

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
