## [Editor Report]

The thalamus is the hub connecting sensory inputs to cortical processing. The elegant study here used 2-photon calcium imaging and behavioral tasks to reveal a role for the posteromedial nucleus of the thalamus in goal directed forepaw behaviors in mice.

---

## [Decision Letter]

**Decision letter after peer review:**

Thank you for submitting your article "Higher order thalamus flexibly encodes correct goal-directed behavior" for consideration by *eLife*. Your article has been reviewed by 3 peer reviewers, and the evaluation has been overseen by a Reviewing Editor and Joshua Gold as the Senior Editor. The reviewers have opted to remain anonymous.

Essential revisions:

1) As is, the optogenetic experiments have some weaknesses. It would be helpful if the authors inhibited during different epochs. For example, their interpretation of what the activity during the reward epoch means is unclear. It would help if authors would inhibit POm during reward- if done in naïve animals, is learning affected? Another potential experiment could be to perform imaging in another circuit terminating in S1. Relatedly, reviewers were concerned that POM targets to output areas beyond somatosensory cortex may contribute to the observed optogenetic inhibition experiments.

2) The main conclusion about the role of POM remains unclear, and additional alternatives should be considered. For example, terms like "behavioral flexibility" are used to describe its purpose, but the connection of this term to POm is not explained.

*Reviewer #1:*

1. Figure 1 – Supp 1 suggests that virus expression was always limited to POm. Drawing borders expressing areas from epifluorescence images is probably very dependent on imaging parameters. The Methods indicate that the authors scaled so that no pixels were saturated. This could mean that there was some weak expression of GCaMP6f or ArchT outside of POm. As I understand it, the authors set exposure/gains by the brightest points in the image. The limited extent of the infection in the figures might just reflect its center, which is brightest, rather than its full extent. If there were GCaMP or ArchT in VPL, some results would need to be reinterpreted.

2. Calcium responses are weaker during the naïve state than the expert state (Figure 1D,E), similar to the start of the reversal training (Figure 4G,H). If POm encodes correct actions, why is there any response at all in naïve mice? Is that not also a sign of stimulus encoding? Might there be another correlate of correctness with regard to the task, such as an expert mouse holding their paw more firmly or still on the stimulating rod? This could alter the effective stimulus or involve different motor signals to POm.

3. The authors are rightly concerned that licking might contribute to POm activity and expend some good effort checking this. The reversal is a good control, but doesn't produce identical POm activity. The other licking analyses, while good, did not completely rule out licking effects. First, lines 110-111 state "…as there was no correlation between licking frequency and POm axonal activity (Figure 1I)", but Figure 1I doesn't seem to support that statement. Second, the authors analyze isolated spontaneous licks, but these probably involve less licking and less overall motion than during a real response.

4. Many figures (Figure 1F, 2B, 3C, 4C) make it apparent that a population of axons respond very early to the stimulus itself. I understand the authors point that many of their analyses show that on average the axons are not strongly modulated by this stimulus, but this is not true of every axon. Either some of these axons are coming from cells outside of POm (see #1) or some POm cells are stimulus driven. In either case, if some axons are strongly stimulus driven, the activity of these axons will correlate with correct choices. The stimulus and correct choices are themselves highly correlated because the animals perform so well. I do not understand how stimulus encoding and choice encoding can be disentangled by either behavior or the two behaviors in comparison. Simple stimulus encoding might be further modulated by arousal or reward expectation that increases with task learning (see #6).

5. I was unable to understand the author's conclusion about what POm is doing. They use terms like "behavioral flexibility" to describe its purpose, but the connection of this term to POm is not explained. Is a role as a flexibility switch really supported? Why does S1 need POm to signal a correct choice? Figure 6 did not seem helpful here. Couldn't S1 just detect the stimulus on its own and transmit consequent signals to wherever they need to be to generate behavior?

6. Arousal or reward expectation may be better explanations than flexibility. Lines 323-324 say that POm activity increased with pupil diameter normally but reversed during reward delivery. Which data support this statement? With regards to pupil, the Results only seem to indicate that there is no difference in diameter between the two conditions (expert and 50% chance) using 3 bins of data. However, I could not find the time windows used for computing these. Pupil is known to be lagged and the timing could be critical.

7. There are other possible interpretations of the results when the authors target POm for optogenetic suppression (around lines 246-248). The effects here are also consistent with preventing tonic and evoked POm activity from reaching lots of target structures other than S1: S2, PPC, motor cortex, dorsolateral striatum, etc. Maybe one of these cannot respond to the stimulus as well and Hits decrease?

8. Line 689. What alerts the mouse that a catch trial is happening? Is there something like an audio cue for onset of stimulus trials and catch trials? If there is no cue, wouldn't mice be in a different behavioral state during catch trials than during stimulus trials? The trial types could differ by more than the presence of the stimulus.

9. Would it be more thorough to zoom in on areas like VPL, set exposures/gains very high, and show that there is no detectable VPL expression or gradient of expression crossing into VPL?

10. The authors indicate that they used video of the paw to exclude trials where the mouse removes the paw entirely from the rod. Why not quantify the paw movements as well and check if the paw is overall moving less in experienced than naïve/switched states? Quantified comparisons of paw stability and calcium are probably also good checks.

11. An analysis that might help would be to check the relationship of lick number/rate and calcium. Third, the authors point out that FA trials have licks but different POm activity (lines 132-134), but the FA and Hit licks may differ in number or frequency. Some check of this is needed.

12. There are many possible ways the authors might address these, and depends on them and the data.

13. Why not just plot the average pupil diameter traces of the two conditions over fairly long time periods?

14. Like 12, the authors may want to deal with these in a variety of ways. On a related note with 7, wouldn't Figure 5E be more informative if latency was broken out by Hits and FAs separately? Related to #1, it would be problematic if the infection had spread into VPL.

*Reviewer #2 (Recommendations for the authors):*

In this manuscript, D LaTerra et al. explored the function of POm neurons during a tactile-based, goal-directed reward behavior. They target POm neurons that project to forepaw S1 and use two-photon ca^2+^imaging in S1 to monitor activity as mice performed a task where forepaw tactile stimulation (200 Hz, 500 ms) predicted a reward if mice licked at a reward port within 1.5 seconds. If mice did not lick, there was a time-out instead of a reward. The authors found that POm-S1 axons showed enhanced responses during the baseline period, the response window after the cue, and during reward delivery. They then showed that a subset of neurons were active during the response window during correct trials when the tactile stimulus served as a cue, but not on catch trials where animals spontaneously licked for a reward.

They then showed that POm axonal activity in S1 increased during the response window for "HIT" trials where animals correctly responded to the tactile stimulus with licking but the activity was less during "MISS" trials where animals did not respond. In order to probe whether this activity in the response window was being driven by motor activity, they designed a suppression task in which animals had to learn to suppress licking in response to the tactile stimulus in order to the receive a reward. POm neurons also showed increased activity during the response window even though action was being suppressed. However, this activity was less than during the action task. Thus, although POm activity is not encoding action, its activity is significantly different during an action-based task than an action suppression one. They then analyzed calclium activity during the training period between the action task and the suppression task in which animals were learning the new contingency and were not performing as experts. In this non-expert context there was not a difference between in POm axonal activity between "HIT" and "MISS" trials.

Lastly, they used ArchT to inhibit POm cell body activity during the tactile stimulus and response window of some trials and showed that they reduced performance during the trials when light was on.

Altogether, this paper provides evidence that POm neurons are not simply encoding sensory information. They are modulated by learning and their activity is correlated to performance in this goal-directed task. However, the actual role of the POm input to S1 is not discernable from the current experiments. Subsets of neurons show significant activity during the response window as well as reward. In addition, the role of this input is different during the switch task than during expert performance. There are a number of outstanding questions, which, if answered, would help to directly define the role of these neurons in this specific paradigm. For instance, the authors record specifically from POm axons in S1. How distinct is this activity from other neurons in the POm? Some POm neurons still show significant activity during MISS trials. Do these neurons have a different function than those that show a preferential response during HIT trials? Does POm activity during the switch task, which has a component of extinction training, differ from when the animals are first learning the action-based task? Likewise, are the same neurons that acquire a response during the initial learning of the action-based task, the same neurons that are responding during the action suppression task?

The authors provide great evidence that POm neurons that project to the S1 do not simply encode sensory information or actions, but are instead signaling during correct performance. However, inhibition of cell bodies did not dramatically effect performance and it is still unclear what role this circuit actually plays in this behavior. Finer-tuned optogenetic experiments and analysis of cell bodies within POm may provide greater details that will help define this circuit's role.

1. Perform optogenetic inhibition during specific epochs of task (response window vs reward) in order to better define this circuit's function.

2. Perform optogenetic inhibition during initial training before learning, to assess if this circuit is necessary for learning this task

3. Calcium imaging was done in POm axons in S1 and was not perfomed in POm itself, yet inhibition was done in cell bodies in POm and the functional role of the projection to S1 was not isolated. Recording cell bodies in POm might help to better characterize sub populations of functional ensembles and how they change during learning. Likewise, inhibiting POm axon terminals in S1 would provide a more nuanced functional assessment of the calcium imaging data presented here.

*Reviewer #3 (Recommendations for the authors):*

In their paper "Higher order thalamus flexibly encodes correct goal-directed behavior", LaTerra et al. investigate the function of projections from the thalamic nucleus POm to primary somatosensory cortex (S1) in the performance of goal-directed behaviors. The authors performed in vivo calcium imaging of POm axons in layer 1 of the forepaw region of S1 (fpS1) to monitor the activity of POm-fpS1 projections while mice performed a tactile detection task. They report that the activity of POm-fpS1 axons on successful ('hit') trials was increased in trained mice relative to naïve mice. Additionally, the authors used an action suppression variant of the task to show that POm-fpS1 axon activity was higher on successful trials over unsuccessful ('miss') trials regardless of the correct motor response required. During transition between task conditions, when mice perform at chance levels, the increase of POm-fpS1 activity during correct trials is no longer seen. Finally, the authors use inhibitory optogenetic tools to suppress POm activity, revealing a modest suppression in behavioral success. The authors conclude from these data that POm-fpS1 axons preferentially "encode and influence correct action selection" during tactile goal-oriented behavior.

This study presents several interesting findings, particularly with respect to the change in activity of POm-fpS1 axons during successful execution of a trained behavior. Additionally, the similarity in responses of POm-fpS1 on both the 'goal-directed action' and 'action suppression' tasks provides convincing evidence that POm-fpS1 activity is not likely to encode the motor response. Overall, these results have important implications for how activity in higher order thalamic nuclei corresponds to learning a sensorimotor behavior, and the authors use several clever experiments to address these questions. Yet, the major claim that POm encodes 'correct performance' should be defined more clearly. As is, there are alternative explanations that could be raised and should be discussed in more depth (Points 1), especially as it relates to any causal role the authors ascribe to POm (Point 2). In addition some clarification as to which types of signals (i.e. frequency of active axons vs. amplitude of signal in the active axons) the authors feel are most informative would be helpful (Point 3).

1) The authors argue that POm activity reflects 'correct task performance' and that the increased activity of POm-fpS1 axons in the response epoch is not due to sensory encoding. An alternative explanation is that POm-fpS1 axons do convey sensory information, and these connections are facilitated with learning – meaning the activity of pathways conveying sensory signals that are correlated with task success could be facilitated with training, and this facilitation could be disrupted during the switching task. In this sense, the activity profiles do not encode 'correct action' per se, but rather represent the sensory responses whose correlation to rewarded action have been reinforced with training (which would also be a very interesting finding). This would be quite distinct from the "cognitive functions" they ascribe to this pathway (line 341). It might have helped to introduce a delay period in between the sensory stimulus and response epoch to try to distinguish responses that encode information about the sensory stimulus from those that might be involved in encoding task performance. However, as is, it is difficult to distinguish between these two scenarios with this data, and thus the interpretations the authors present could be rephrased with alternatives discussed in more depth.

2) Similarly, while the authors attempt to establish a causal role for POm in task performance by optogenetically inhibiting POm during the response epoch, the results are also consistent with a deficit in sensory processing, and cannot be interpreted strictly as a disruption of the encoding of 'correct action' task performance signals. Furthermore, these perturbation studies do not demonstrate that the POm-fpS1 projections they are studying are implicated in the modest behavioral deficits. As the authors state, POm projects to many targets (lines 63-66), and similar sensory-based, goal-directed behaviors do not require S1 (lines 302-305). In light of these points, some of the statements ascribing a causal role for these projections in task success could be rephrased (e.g. line 33 "to encode and influence correct action selection", line 252 "a direct influence", line 340 "plays an active role during correct performance").

3) Event amplitude and probability were both quantified, but were not consistently reported throughout the manuscript and figures. For example, Figure 1 reports both probability and amplitude (Figure 1G and H), whereas Figure 2 only reports probability. Thus, it was not always clear as to whether the authors were ascribing biological significance to one or both of these measures, given that in some cases differences were found in one and not the other, and which of the measures were reported was occasionally switched. It would be helpful for the authors to clarify the significance they assign to each measure, and report both measures side by side for all experiments if they interpret them both as relevant.

4. It was unclear why the authors did not attempt to use deconvolution and report spike probabilities, especially when considering the kinetics of GCaMP6f and the results presented in Figure 4, where event amplitude and event probability changed in opposing directions, which could reflect a change in burst firing since spikes in short high frequency bursts can appear as a single large amplitude event compared to single spikes. The authors could consider performing further analysis or discuss the caveats of analyzing the Ca signal across these large time windows.

5. It would be helpful to clarify the basis upon which boutons or ROIs were excluded when determining the 'axon subset' of ROIs. How did Ca event probability, event amplitude, and event duration compare between ROIs that were assessed as being from the same axon? It was unclear what was deemed as 'similar activity profiles' for the exclusion of ROIs. It might help to include an additional figure supplement to Figure 1 showing the ROI correlations and exclusion criteria with images showing 'similar' or 'dissimilar' ROIs marked on an example field of view.

6. The heterogeneity of POm axons was briefly shown (Figure 1) but not discussed or explored in depth. It was unclear how the authors interpret the observation that a subset of POm-fpS1 axons showed a larger increase in the reward epoch compared to the stimulus and response epochs. While this diversity in responses could be relevant to their claim that POm-fpS1 axons encode task performance, the authors did not perform experiments inhibiting POm during the reward epoch, leaving unclear the interpretation of what the reward epoch responses might mean. More discussion of the interpretation of POm-fpS1 activity during the reward epoch would be helpful, given that several sections of the Results are dedicated to this point. Also, a clearer descriptions of row sorting in the figures (e.g. Figure 2B) would enable more direct comparison of activity of the same axon across different trial types.

[Editors’ note: the authors submitted for reconsideration following the decision after peer review. What follows is the decision letter after the second round of review.]

Thank you for resubmitting the paper entitled "Higher order thalamus flexibly encodes correct goal-directed behavior" for further consideration by *eLife*. Your revised article has been evaluated by a Senior Editor and a Reviewing Editor. We are sorry to say that we have decided that this submission will not be considered further for publication by *eLife*.

Although reviewers acknowledge improvement in the clarity of the manuscript, key experiments that were requested were not performed, such as inhibiting during different epochs or in naïve animals. Moreover, a major issue that was raised in the first round was to clarify what function the authors are ascribing to POm. Although the reviewers acknowledge improvement in language, the reviewers all felt that it is still not clear why Pom is needed to signal correct performance, and the data do not exactly support the conclusion that POm=correct.

---

## [Author Response]

Essential revisions:1) As is, the optogenetic experiments have some weaknesses. It would be helpful if the authors inhibited during different epochs. For example, their interpretation of what the activity during the reward epoch means is unclear. It would help if authors would inhibit POm during reward- if done in naïve animals, is learning affected? Another potential experiment could be to perform imaging in another circuit terminating in S1. Relatedly, reviewers were concerned that POM targets to output areas beyond somatosensory cortex may contribute to the observed optogenetic inhibition experiments.

Due to the COVID pandemic, revising this manuscript and performing additional experiments and analysis has been extremely slow. In the past 1.5 years, Melbourne, Australia has endured 260+ days of lockdown which has been reportably the strictest and longest in the world. This has affected our ability to perform the requested experiments as the laboratory was shut down for months on end, mice were culled and experiments ceased.

Despite the delays, we were able to perform the requested additional experiments:

1) We have now performed additional experiments where we inhibit POm during learning of the tactile goal-directed behavior. Here, we found learning was severely disrupted and mice took approximately two-times longer to learn the task. This data is now included in new Figure 6.

2) To assess the role of POm activity during the reward delivery, we photo-inactivated the POm during the reward epoch in expert mice. Here we found no influence of POm photo-inactivation during reward delivery on mouse performance. This result, which corresponds to the lower rates in the activity of POm axons within S1 during reward delivery, is now included in Figure 5.

3) We attempted to use pathway specific DREADDs to specifically target the POm input that synapses in S1. Using dual injection of retrograde cre into S1, and cre-dependent DREADDs into POm, this strategy should have enabled us to specifically inactivate the S1-targeting POm pathway during the task. Unfortunately, after much effort, we were unable to confirm specificity of the labelling and therefore were ultimately unable to persist with this dataset.

2) The main conclusion about the role of POM remains unclear, and additional alternatives should be considered. For example, terms like "behavioral flexibility" are used to describe its purpose, but the connection of this term to POm is not explained.

In the revised manuscript, the main conclusions about the POm are now clearly stated and the use of the term ‘behavioral flexibility’ has been limited to improve the clarity of the findings.

Reviewer #1 (Recommendations for the authors):1. Figure 1 – Supp 1 suggests that virus expression was always limited to POm. Drawing borders expressing areas from epifluorescence images is probably very dependent on imaging parameters. The Methods indicate that the authors scaled so that no pixels were saturated. This could mean that there was some weak expression of GCaMP6f or ArchT outside of POm. As I understand it, the authors set exposure/gains by the brightest points in the image. The limited extent of the infection in the figures might just reflect its center, which is brightest, rather than its full extent. If there were GCaMP or ArchT in VPL, some results would need to be reinterpreted.

We agree with the reviewer that the determined expression areas are dependent on imaging parameters, however, we are confident that the virus expression used for analysis in this study are confined to the POm. In this study, our analysis of targeting of POm is three-fold. First, we optimized the volume of virus loaded to the minimum necessary to observe POm projections in S1 (a single targeted injection of 60 nl). Second, we analyzed the fluorescence spread using fluorescence microscopy after every experiment. We set exposure to use the full dynamic range of the image as previously described (Gambino et al., 2014). Occasionally, the virus spread to the adjacent VPM nucleus and this was easily recognizable by the characteristic VPM projections with the barrels of the barrel cortex. These animals were excluded from this study and not further analyzed. The VPL nucleus is located further caudally in respect to the VPM and again, we were able to identify if the virus has spread to this nucleus via posthoc fluorescence microscopy. These animals were excluded from this study and not further analyzed. We note that our stereotaxic injections were not flawless and the virus occasionally spread along the injection pipette track and into high-order visual thalamic nuclei LP and LD, superficial to POm. This is shown in Figure 1. These two nuclei, however, do not target S1 (Kamishina et al., 2009; van Groen and Wyss, 1992) and were therefore not imaged within our study. Third, we analyze the projection profile in the forepaw area of S1 to ensure that it corresponds to the projection profile of POm and not VPL. If there is fluorescence in nontargeted areas, then the experiments were excluded from analysis.

An additional degree of precision is offered by our imaging and optogenetic strategy. Calcium imaging was performed in layer 1 which is primarily targeted by Pom (Meyer et al., 2010), and not VPL which primarily targets layer 4. Therefore, spillover into VPL would not influence our imaging results as we only image axons in layer 1 which is targeted by Pom. Furthermore, during the optogenetic experiments, the fiber optic was targeted to the Pom (not the VPL), thus providing a secondary Pom localization of the photo-inhibited region. This is now discussed in the revised manuscript.

2. Calcium responses are weaker during the naïve state than the expert state (Figure 1D,E), similar to the start of the reversal training (Figure 4G,H). If Pom encodes correct actions, why is there any response at all in naïve mice? Is that not also a sign of stimulus encoding? Might there be another correlate of correctness with regard to the task, such as an expert mouse holding their paw more firmly or still on the stimulating rod? This could alter the effective stimulus or involve different motor signals to Pom.

We agree with the reviewer that the Pom is encoding the stimulus in the naïve state. This is evident in our study, and others, which show that the Pom increases activity during sensory input in naïve mice. In the expert state, stimulus encoding may also be performed by a subset of Pom axons, and we now discuss this in the revised manuscript. However, the results from our study, and other studies of different pathways, illustrates that overall sensory encoding reorganizes with learning (Chen et al., 2015; Reinert et al., 2021), with smaller sensory-evoked responses during active sensory-based behavior (Sachidhanandam et al., 2013).

Overall, our findings show that, during task performance, there is a significant increase in the Pom activity during the stimulus/response epoch which is dependent on the behavioral performance (HIT, MISS). This is not due to licking motion as there was similar Pom activity during the action and suppression tasks which involved licking and not licking for reward (Figure 3). Furthermore, all experiments were monitored online via a behavioral camera to examine the location of the forepaw on the stimulus during all trials, and trials where the paw was not clearly resting on the stimulating rod, or where excessive motion was detected were excluded from analysis. This is now discussed in the revised manuscript.

3. The authors are rightly concerned that licking might contribute to Pom activity and expend some good effort checking this. The reversal is a good control, but doesn’t produce identical Pom activity. The other licking analyses, while good, did not completely rule out licking effects. First, lines 110-111 state “…as there was no correlation between licking frequency and Pom axonal activity (Figure 1I)”, but Figure 1I doesn’t seem to support that statement. Second, the authors analyze isolated spontaneous licks, but these probably involve less licking and less overall motion than during a real response.

We thank the reviewer for acknowledging the effort we made to assess the influence of licking behavior on Pom axonal activity. We now include a more direct analysis in the revised manuscript illustrating the relationship between the licking response and Pom activity. This analysis shows there is no correlation between licking and Pom axonal activity (linear regression, p = 0.923), further suggesting that Pom axonal activity is not simply due to licking behavior.

4. Many figures (Figure 1F, 2B, 3C, 4C) make it apparent that a population of axons respond very early to the stimulus itself. I understand the authors point that many of their analyses show that on average the axons are not strongly modulated by this stimulus, but this is not true of every axon. Either some of these axons are coming from cells outside of Pom (see #1) or some Pom cells are stimulus driven. In either case, if some axons are strongly stimulus driven, the activity of these axons will correlate with correct choices. The stimulus and correct choices are themselves highly correlated because the animals perform so well. I do not understand how stimulus encoding and choice encoding can be disentangled by either behavior or the two behaviors in comparison. Simple stimulus encoding might be further modulated by arousal or reward expectation that increases with task learning (see #6).

We agree with the reviewer that individual Pom axons are heterogenous and a subset of axons may respond to the sensory stimulus during the behavior. We have now performed analysis on a subset of Pom axons (n = 107 axons) where the ca^2+^ transients have been further subcategorized as either occurring during the stimulus and response epochs during the goaldirected task. Here, only a small portion of the axons (6 %) were active only during the stimulus, therefore, although we agree that there is a subset of axons that are stimulus driven, they are a minority during expert behavior. This is information is now included in the revised manuscript.

In this study, we attempted to disentangle stimulus and choice encoding by comparing the Pom axonal activity with the different behavioral performance (HIT or MISS). Here, the same stimulus is always presented (tactile, 200 Hz), however, the mouse response differs. Despite receiving the same tactile stimulus, Pom signaling in forepaw S1 is significantly increased during correct HIT trials compared with MISS trials in both the action and suppression task. Therefore, combined with the small number of axons that were stimulus driven, our results illustrate that Pom axonal activity is predominantly encoding behavioral information in this task.

We agree that simple stimulus encoding might be further modulated by arousal or reward expectation that increases with task learning. In our study, the increase in Pom activity during HIT behavior was not due to elevated task engagement as, despite similar levels of arousal (Figure 4B), Pom activity in expert mice differed in comparison to chance performance (switch behavior; Figure 4E,F). This is now discussed in the revised manuscript.

5. I was unable to understand the author’s conclusion about what Pom is doing. They use terms like “behavioral flexibility” to describe its purpose, but the connection of this term to Pom is not explained. Is a role as a flexibility switch really supported? Why does S1 need Pom to signal a correct choice? Figure 6 did not seem helpful here. Couldn’t S1 just detect the stimulus on its own and transmit consequent signals to wherever they need to be to generate behavior?

We have now revised the manuscript to improve the clarity of our conclusions and have removed (old) Figure 6 as it wasn’t helpful. Overall, our findings suggest that the Pom provides input to S1 which preferentially encodes correct (HIT) performance during goal-directed behavior. Specifically, the POm is primarily active during the behavioral response epoch, and not reward delivery. Furthermore, photo-inactivation of the POm during learning illustrates the POm influences the rate of learning of the goal-directed task.

If S1 simply detected the stimulus on its own and transmitted a consequent signals to generate behavior, then important modulatory processes required during behavior would not be possible. Along with other feedback projections, the POm can provide instant, and dynamic, feedback information to S1 to directly alter cortical signals. Specifically, POm targets the upper layers of the cortex, whereas external sensory information targets the layer 4 input layer. At the level of a single pyramidal neuron, this means POm input lands on the tuft dendrites whereas external sensory information lands on the proximal basal dendrites. This segregation of input provides a great cellular mechanism for increasing the computational capabilities and modulation of neurons – which would be lost if S1 simply detected the stimulus and transmitted consequent signals to generate behavior.

We have now performed additional experiments where we photo-inactivate the POm during learning (new Figure 6) and also during the reward epoch in expert mice (new panels in Figure 5). These additional experiments have shed more light on the role and influence of the POm during goal-directed behavior. Here, we illustrate the POm plays an important role during the response, and not reward, epoch in correct (HIT) behavior and during learning.

6. Arousal or reward expectation may be better explanations than flexibility. Lines 323-324 say that POm activity increased with pupil diameter normally but reversed during reward delivery. Which data support this statement? With regards to pupil, the Results only seem to indicate that there is no difference in diameter between the two conditions (expert and 50% chance) using 3 bins of data. However, I could not find the time windows used for computing these. Pupil is known to be lagged and the timing could be critical.

We have now revised the manuscript to further expand the reporting of our findings. The statement that ‘POm activity increased with pupil diameter normally but reversed during reward delivery’ stems from data illustrated in Figure 1I and 3B. For space and flow of the manuscript, we were not able to show them on the same graph as Author response image 1. Here, you can see that during reward (blue), POm activity decreased compared to response (green) whereas the pupil diameter was maximum during reward delivery. We now include more information in the methods regarding pupil tracking (see Data analysis and statistical methods; Pupil tracking).

**Author response image 1. sa2fig1:** 

7. There are other possible interpretations of the results when the authors target POm for optogenetic suppression (around lines 246-248). The effects here are also consistent with preventing tonic and evoked POm activity from reaching lots of target structures other than S1: S2, PPC, motor cortex, dorsolateral striatum, etc. Maybe one of these cannot respond to the stimulus as well and Hits decrease?

We agree with the reviewer that there are other possible interpretations of the optogenetic suppression experiments. We now discuss this in the revised manuscript.

8. Line 689. What alerts the mouse that a catch trial is happening? Is there something like an audio cue for onset of stimulus trials and catch trials? If there is no cue, wouldn't mice be in a different behavioral state during catch trials than during stimulus trials? The trial types could differ by more than the presence of the stimulus.

There is broadband noise during the trial that acts as a cue. This is detailed in the methods and text.

9. Would it be more thorough to zoom in on areas like VPL, set exposures/gains very high, and show that there is no detectable VPL expression or gradient of expression crossing into VPL?

We are confident that minimal VPL expression is not influencing our imaging and photosuppression results due to imaging axons only in the upper cortical layers and targeting the POm with the fiber optic respectively. This was confirmed in all recordings.

10. The authors indicate that they used video of the paw to exclude trials where the mouse removes the paw entirely from the rod. Why not quantify the paw movements as well and check if the paw is overall moving less in experienced than naïve/switched states? Quantified comparisons of paw stability and calcium are probably also good checks.

In this study, we used an infrared webcam to monitor body movement. Using this, we were able to exclude any trials where the stimulus was not delivered to the paw, however, due to low resolution, the video quality does not allow us to perform detailed tracking of the paw. We now acknowledge this limitation in the methods. We did, however, analyze at great length the licking behavior and ensured there was not a correlation between licking and POm activity.

11. An analysis that might help would be to check the relationship of lick number/rate and calcium. Third, the authors point out that FA trials have licks but different POm activity (lines 132-134), but the FA and Hit licks may differ in number or frequency. Some check of this is needed.

We have now analyzed the lick rate and correlated it with POm activity. In brief, there is no correlation between licking rate and POm activity and we now include this information in the revised manuscript.

We have also analyzed evoked lick rates in FA and HIT trials. There was no significant difference between FA and HIT lick rates (p = 0.203; n = 9 mice).

12. There are many possible ways the authors might address these, and depends on them and the data.

We thank the reviewer for these suggestions and we have attempted to address all points vigorously.

13. Why not just plot the average pupil diameter traces of the two conditions over fairly long time periods?

We apologize, but we are not quite sure what the reviewer is suggesting. In our study, tracking pupil dilation was used to test whether the arousal state of the mouse was similar in the different behavioral epochs. Using the analysis we performed, we were able to show that arousal was similar in the different tasks.

14. Like 12, the authors may want to deal with these in a variety of ways. On a related note with 7, wouldn't Figure 5E be more informative if latency was broken out by Hits and FAs separately? Related to #1, it would be problematic if the infection had spread into VPL.

In Figure 5D and Figure 5 —figure supplement 2 (old Figure 5E), we focus on the licking latency only during correct HIT performance in mice injected with Archaeorhopsin (top) and GFP (bottom). This serves as an important control to illustrate that LED itself does not influence licking behavior/latency.

In the optogenetic photo-inhibition experiments, targeting of the fiber optic canula to the POm and viral expression was confirmed after every experiment. Weak expression of ArchT outside of the POm would therefore also have minimal impact on our findings. This is now detailed in the revised manuscript.

Reviewer #2 (Recommendations for the authors):In this manuscript, D LaTerra et al. explored the function of POm neurons during a tactile-based, goal-directed reward behavior. They target POm neurons that project to forepaw S1 and use two-photon ca^2+^imaging in S1 to monitor activity as mice performed a task where forepaw tactile stimulation (200 Hz, 500 ms) predicted a reward if mice licked at a reward port within 1.5 seconds. If mice did not lick, there was a time-out instead of a reward. The authors found that POm-S1 axons showed enhanced responses during the baseline period, the response window after the cue, and during reward delivery. They then showed that a subset of neurons were active during the response window during correct trials when the tactile stimulus served as a cue, but not on catch trials where animals spontaneously licked for a reward.They then showed that POm axonal activity in S1 increased during the response window for "HIT" trials where animals correctly responded to the tactile stimulus with licking but the activity was less during "MISS" trials where animals did not respond. In order to probe whether this activity in the response window was being driven by motor activity, they designed a suppression task in which animals had to learn to suppress licking in response to the tactile stimulus in order to the receive a reward. POm neurons also showed increased activity during the response window even though action was being suppressed. However, this activity was less than during the action task. Thus, although POm activity is not encoding action, its activity is significantly different during an action-based task than an action suppression one. They then analyzed calclium activity during the training period between the action task and the suppression task in which animals were learning the new contingency and were not performing as experts. In this non-expert context there was not a difference between in POm axonal activity between "HIT" and "MISS" trials.Lastly, they used ArchT to inhibit POm cell body activity during the tactile stimulus and response window of some trials and showed that they reduced performance during the trials when light was on.Altogether, this paper provides evidence that POm neurons are not simply encoding sensory information. They are modulated by learning and their activity is correlated to performance in this goal-directed task. However, the actual role of the POm input to S1 is not discernable from the current experiments. Subsets of neurons show significant activity during the response window as well as reward. In addition, the role of this input is different during the switch task than during expert performance. There are a number of outstanding questions, which, if answered, would help to directly define the role of these neurons in this specific paradigm. For instance, the authors record specifically from POm axons in S1. How distinct is this activity from other neurons in the POm? Some POm neurons still show significant activity during MISS trials. Do these neurons have a different function than those that show a preferential response during HIT trials? Does POm activity during the switch task, which has a component of extinction training, differ from when the animals are first learning the action-based task? Likewise, are the same neurons that acquire a response during the initial learning of the action-based task, the same neurons that are responding during the action suppression task?The authors provide great evidence that POm neurons that project to the S1 do not simply encode sensory information or actions, but are instead signaling during correct performance. However, inhibition of cell bodies did not dramatically effect performance and it is still unclear what role this circuit actually plays in this behavior. Finer-tuned optogenetic experiments and analysis of cell bodies within POm may provide greater details that will help define this circuit's role.

We thank the reviewer for their comments. We have now revised the manuscript to clearly state the role of the POm during the goal-directed behavioral tasks used in this study. We have provided more information regarding the range of activity patterns in POm axons within S1.

The POm contains a heterogenous population of neurons and since it projects to multiple cortical and subcortical regions, the activity of POm axonal projections in S1 may indeed be different to other projection targets.

The activity of POm axons during MISS behavior may have a different function than those that show a preferential response during HIT trials, however, this evoked rate is not significantly different to baseline and therefore is hard to differentiate from spontaneous activity (see Figure 2). Furthermore, the evoked rate of POm activity during the switch task is not significantly different compared to naïve mice (p = 0.159; Kruskal-Wallis test). This information is now included in the manuscript.

It is unknown whether the same neurons that acquire a response during the initial learning of the action-based task are the same neurons that are responding during the action suppression task as we were unable to conclusively determine whether or not the same POm axons were imaged in the different protocols.

1. Perform optogenetic inhibition during specific epochs of task (response window vs reward) in order to better define this circuit's function.

In this study, we inhibited the POm during the stimulus and response epoch as this is when the POm was most active. We now perform experiments where we restrict optogenetic inhibition of the POm to the reward window. As shown in new Figure 5E and F, photo-inactivating the POm during the reward window did not alter behavioral performance. In combination with the finding that POm axons within S1 are considerably less active during the reward epoch compared with the response epoch, our findings suggest that the POm does not predominantly play a role in reward encoding and behavior. In the revised manuscript, we now include more details about the limitations of using Archaerhodopsin to optogenetically silence the POm.

2. Perform optogenetic inhibition during initial training before learning, to assess if this circuit is necessary for learning this task

We have now performed new experiments where we optogenetically inhibit the POm during learning of the goal-directed task. Here, POm photo-inactivation decreased the rate of learning the tactile goal-directed task, significantly increasing the number of sessions required to reach expert (>80 % correct) performance (p = 0.0036). These findings illustrate that the POm is involved in learning this task and this information is now included as a new Figure 6 in the revised manuscript.

3. Calcium imaging was done in POm axons in S1 and was not perfomed in POm itself, yet inhibition was done in cell bodies in POm and the functional role of the projection to S1 was not isolated. Recording cell bodies in POm might help to better characterize sub populations of functional ensembles and how they change during learning. Likewise, inhibiting POm axon terminals in S1 would provide a more nuanced functional assessment of the calcium imaging data presented here.

We agree with the reviewer that it would be of great to record from POm cell bodies during behavior, however, in this study, we were particularly interested in recording the information transferred from the POm to S1. Since the POm projects to different brain regions, here we isolated the POm projections that target S1 by specifically recording from POm axons within S1. Furthermore, it is difficult to perform calcium imaging to the depth of the POm without perturbing the cortex, which may influence sensory-based behavior. Likewise, another technique used to record neural activity is whole-cell patch clamp electrophysiology, however it is difficult to perform recordings during behavior due to the movement involved in the behavioral response.

We attempted to inhibit Pom axon terminals in S1, however, unfortunately, the attempts were unsuccessful. Firstly, we used the genetically expressed chloride channel, gtACR, to inhibit axonal terminals. However, our control experiments illustrated that photo-activation of gtACR in Pom axonal terminals in S1 results in excitation, and not inhibition (Malyshev *et al.,* 2017). Although not optimal, our supplementary control experiments (Figure 5 —figure supplement 1) clearly show that we were able to photo-inhibit Pom neurons expressing Archaerhodopsin when LED was restricted to the cell body.

Reviewer #3 (Recommendations for the authors):In their paper “Higher order thalamus flexibly encodes correct goal-directed behavior”, LaTerra et al. investigate the function of projections from the thalamic nucleus Pom to primary somatosensory cortex (S1) in the performance of goal-directed behaviors. The authors performed in vivo calcium imaging of Pom axons in layer 1 of the forepaw region of S1 (fpS1) to monitor the activity of Pom-fpS1 projections while mice performed a tactile detection task. They report that the activity of Pom-fpS1 axons on successful (‘hit’) trials was increased in trained mice relative to naïve mice. Additionally, the authors used an action suppression variant of the task to show that Pom-fpS1 axon activity was higher on successful trials over unsuccessful (‘miss’) trials regardless of the correct motor response required. During transition between task conditions, when mice perform at chance levels, the increase of Pom-fpS1 activity during correct trials is no longer seen. Finally, the authors use inhibitory optogenetic tools to suppress Pom activity, revealing a modest suppression in behavioral success. The authors conclude from these data that Pom-fpS1 axons preferentially “encode and influence correct action selection” during tactile goal-oriented behavior.This study presents several interesting findings, particularly with respect to the change in activity of Pom-fpS1 axons during successful execution of a trained behavior. Additionally, the similarity in responses of POm-fpS1 on both the ‘goal-directed action’ and ‘action suppression’ tasks provides convincing evidence that Pom-fpS1 activity is not likely to encode the motor response. Overall, these results have important implications for how activity in higher order thalamic nuclei corresponds to learning a sensorimotor behavior, and the authors use several clever experiments to address these questions. Yet, the major claim that Pom encodes ‘correct performance’ should be defined more clearly. As is, there are alternative explanations that could be raised and should be discussed in more depth (Points 1), especially as it relates to any causal role the authors ascribe to Pom (Point 2). In addition some clarification as to which types of signals (i.e. frequency of active axons vs. amplitude of signal in the active axons) the authors feel are most informative would be helpful (Point 3).

We thank the reviewer for their helpful comments and assessment of our study. We have now addressed all comments and revised the manuscript accordingly.

1) The authors argue that Pom activity reflects ‘correct task performance’ and that the increased activity of Pom-fpS1 axons in the response epoch is not due to sensory encoding. An alternative explanation is that Pom-fpS1 axons do convey sensory information, and these connections are facilitated with learning – meaning the activity of pathways conveying sensory signals that are correlated with task success could be facilitated with training, and this facilitation could be disrupted during the switching task. In this sense, the activity profiles do not encode 'correct action' per se, but rather represent the sensory responses whose correlation to rewarded action have been reinforced with training (which would also be a very interesting finding). This would be quite distinct from the “cognitive functions” they ascribe to this pathway (line 341). It might have helped to introduce a delay period in between the sensory stimulus and response epoch to try to distinguish responses that encode information about the sensory stimulus from those that might be involved in encoding task performance. However, as is, it is difficult to distinguish between these two scenarios with this data, and thus the interpretations the authors present could be rephrased with alternatives discussed in more depth.

We agree that it would have been beneficial to separate the stimulus from the response period in the behavioral paradigm. However, to avoid engaging working memory, we did not wish to enforce a delay period in this study.

Furthermore, we have now performed analysis on a subset of Pom axons (n = 107 axons) where the ca^2+^ transients have been further sub-categorized as either occurring during the stimulus and response epochs during the goal-directed task. Here, only a small portion of the axons (6 %) were active only during the stimulus, therefore, although we agree that there is a subset of axons that are stimulus driven, they are a minority during expert behavior. This is information is now included in the revised manuscript.

In this study, we attempted to disentangle stimulus and choice encoding by comparing the Pom axonal activity with the different behavioral performance (HIT or MISS). Here, the same stimulus is always presented (tactile, 200 Hz), however, the mouse response differs. Despite receiving the same tactile stimulus, Pom signaling in forepaw S1 is significantly increased during correct HIT trials compared with MISS trials in both the action and suppression task. Therefore, combined with the small number of axons that were stimulus driven, our results illustrate that Pom axonal activity is predominantly encoding behavioral information in this task.

2) Similarly, while the authors attempt to establish a causal role for Pom in task performance by optogenetically inhibiting Pom during the response epoch, the results are also consistent with a deficit in sensory processing, and cannot be interpreted strictly as a disruption of the encoding of ‘correct action’ task performance signals. Furthermore, these perturbation studies do not demonstrate that the Pom-fpS1 projections they are studying are implicated in the modest behavioral deficits. As the authors state, Pom projects to many targets (lines 63-66), and similar sensory-based, goal-directed behaviors do not require S1 (lines 302-305). In light of these points, some of the statements ascribing a causal role for these projections in task success could be rephrased (e.g. line 33 “to encode and influence correct action selection”, line 252 “a direct influence”, line 340 “plays an active role during correct performance”).

We agree that the decrease in correct performance during optogenetic inhibition of Pom cell bodies may also be explained by a deficit in sensory processing. However, in this study, we went to great lengths to illustrate that the Pom is encoding correct action, and not sensory information (detailed in response to 1). This is further expanded upon in the revised manuscript. We also agree that the perturbation studies do not directly demonstrate that the Pom to S1 projections are driving the behavioral deficits. We attempted to use pathway specific DREADDs to specifically inactivate the Pom input that targets S1. Using dual injection of retrograde-cre into S1, and cre-dependent DREADDs into Pom, this strategy should have enabled us to specifically inactivate the S1-targeting Pom pathway during the task. Unfortunately, after much effort, we were unable to confirm specificity of the labelling and therefore were ultimately unable to persist with this dataset. We therefore only refer to the Pom itself when discussing the influence on behavior and we have now revised the manuscript accordingly.

3) Event amplitude and probability were both quantified, but were not consistently reported throughout the manuscript and figures. For example, Figure 1 reports both probability and amplitude (Figure 1G and H), whereas Figure 2 only reports probability. Thus, it was not always clear as to whether the authors were ascribing biological significance to one or both of these measures, given that in some cases differences were found in one and not the other, and which of the measures were reported was occasionally switched. It would be helpful for the authors to clarify the significance they assign to each measure, and report both measures side by side for all experiments if they interpret them both as relevant.

We thank the reviewer for this observation and have now included a statement discussing the reporting of Ca^2+^ transient probability and/or amplitude in the methods. Throughout the Figures, we typically illustrated probability of an evoked transient as this is a reliable measure which was dramatically altered within this study. We now report the Ca^2+^ transient peak amplitudes during HIT and MISS trials for direct comparison of both measures (Figure 2).

4. It was unclear why the authors did not attempt to use deconvolution and report spike probabilities, especially when considering the kinetics of GcaMP6f and the results presented in Figure 4, where event amplitude and event probability changed in opposing directions, which could reflect a change in burst firing since spikes in short high frequency bursts can appear as a single large amplitude event compared to single spikes. The authors could consider performing further analysis or discuss the caveats of analyzing the Ca signal across these large time windows.

We agree that it would be informative to deconvolve our calcium transients, however, since we don’t know the ground truth, we would not be confident in reporting convoluted spike probabilities. It is for this reason that we report both evoked rates and amplitudes, and we now include a statement on the caveats of analyzing the calcium signal across large time windows in the revised manuscript.

5. It would be helpful to clarify the basis upon which boutons or ROIs were excluded when determining the ‘axon subset’ of ROIs. How did Ca event probability, event amplitude, and event duration compare between ROIs that were assessed as being from the same axon? It was unclear what was deemed as ‘similar activity profiles’ for the exclusion of ROIs. It might help to include an additional figure supplement to Figure 1 showing the ROI correlations and exclusion criteria with images showing ‘similar’ or ‘dissimilar’ ROIs marked on an example field of view.

In this study, ROIs were excluded if they had greater than 95% of same temporal activity pattern (evoked probability) as a neighboring ROI in the same FOV. In other words, the activity patterns were essentially identical, but we allowed a small window for varying signal-to-noise ratios in the different boutons. We now include this additional information in the methods, and include an additional Figure Supplement to Figure 1 to clearly illustrate our exclusion criteria (Figure 1 —figure supplement 4).

6. The heterogeneity of Pom axons was briefly shown (Figure 1) but not discussed or explored in depth. It was unclear how the authors interpret the observation that a subset of Pom-fpS1 axons showed a larger increase in the reward epoch compared to the stimulus and response epochs. While this diversity in responses could be relevant to their claim that Pom-fpS1 axons encode task performance, the authors did not perform experiments inhibiting Pom during the reward epoch, leaving unclear the interpretation of what the reward epoch responses might mean. More discussion of the interpretation of Pom-fpS1 activity during the reward epoch would be helpful, given that several sections of the Results are dedicated to this point. Also, a clearer descriptions of row sorting in the figures (e.g. Figure 2B) would enable more direct comparison of activity of the same axon across different trial types.

We have now performed new experiments where we photo-inactivate the Pom during the reward epoch in the tactile goal-directed task. Here, we see no change in the task performance (see new panels in Figure 5). Together with the finding that Pom axons within S1 are less active during the reward epoch compared with the response epoch, our findings suggest that the Pom does not predominantly signal reward during expert behavior.

We now provide more detail in the Figure captions and methods regarding row sorting of the ca^2+^ activity patterns (heatmaps) to improve clarity and interpretation.

[Editors’ note: what follows is the authors’ response to the second round of review.]

Although reviewers acknowledge improvement in the clarity of the manuscript, key experiments that were requested were not performed, such as inhibiting during different epochs or in naïve animals. Moreover, a major issue that was raised in the first round was to clarify what function the authors are ascribing to POm. Although the reviewers acknowledge improvement in language, the reviewers all felt that it is still not clear why Pom is needed to signal correct performance, and the data do not exactly support the conclusion that POm=correct.

This study directly investigates the role of higher order somatosensory thalamus (POm) in the forepaw area of the primary somatosensory cortex during sensory-based goal-directed behavior, and highlights the important role this thalamic input has on behavioral performance and learning. In this new revised manuscript, we have now addressed all of the comments/suggestions from the reviewers and made the following major changes:

1) Optogenetically inhibit the POm during learning of the tactile goal-directed behavior. Here, we found learning was severely disrupted and mice took approximately two-times longer to learn the task when the POm was photo-inactivated. This data is now included in new Figure 6.

2) To assess the role of POm activity during the reward delivery, we photo-inactivated the POm during the reward epoch in expert mice. Here we found no influence of POm photo-inactivation during reward delivery on mouse performance. We also performed control experiments where mice were instead injected with GFP. This result, which corresponds to the lower rates in the activity of POm axons within S1 during reward delivery, is now included in Figure 5.

3) Included an additional Figure Supplement to Figure 1 to clearly illustrate our POm axonal inclusion and exclusion criteria (Figure 1 —figure supplement 4).

4) Assessed the efficacy of POm axonal photoinhibition in vitro*.* This data is now included as a new panel in Figure 5 —figure supplement 1.

5) Included a more direct analysis in the revised manuscript illustrating the relationship between the licking response and POm activity. This analysis shows there is no correlation between licking and POm axonal activity, further suggesting that POm axonal activity is not simply due to licking behavior.

6) Report the POm axonal ca^2+^ transient peak amplitudes during HIT and MISS trials for direct comparison (Figure 2).

7) Compared the incorrect trials across the three task types and included this information in the revised manuscript.

8) Overall, the manuscript has been revised and rewritten to improve the reporting of the results.

Taken together, these new additions to the manuscript have strengthened the findings that the POm encodes and influences correct action selection in learnt behavior, and additional experiments have highlighted an important role of the POm during learning.